# Computing associators of endomorphism fusion categories

Daniel Barter,[1, *] Jacob C. Bridgeman,[2, †] and Ramona Wolf[3, ‡]

[1]*Lawrence Berkeley National Laboratory, Berkeley, California, United States*
[2]*Perimeter Institute for Theoretical Physics, Waterloo, Ontario, Canada*
[3]*Institute for Theoretical Physics, ETH Zürich, Zurich, Switzerland*

(Dated: July 28, 2022)

Many applications of fusion categories, particularly in physics, require the associators or $F$-symbols to be known explicitly. Finding these matrices typically involves solving vast systems of coupled polynomial equations in large numbers of variables. In this work, we present an algorithm that allows associator data for some category with unknown associator to be computed from a Morita equivalent category with known data. Given a module category over the latter, we utilize the representation theory of a module tube category, built from the known data, to compute this unknown associator data. When the input category is unitary, we discuss how to ensure the obtained data is also unitary.

We provide several worked examples to illustrate this algorithm. In addition, we include several Mathematica files showing how the algorithm can be used to compute the data for the Haagerup category $\mathcal{H}_1$, whose data was previously unknown.

## I. INTRODUCTION

To perform calculations within fusion categories that involve working in a specific basis, it is necessary that the associators, also called the $F$-symbols, are known. In particular, they are a crucial ingredient in the construction of physical models such as one- or two-dimensional lattice models [1, 2].

The $F$-symbols can be obtained by solving the pentagon equations (see, for example, Ref. [3]), which amounts to solving a system of multivariate polynomial equations up to third order. The number of variables, and equations they must satisfy, grows rapidly with the number of simple objects in the category, meaning that solving this problem quickly becomes impractical. In fact, the growth in complexity is so rapid that few associators are known for fusion categories with more than six simple objects. The challenge of finding $F$-symbols becomes even more significant for categories with multiplicities, as the number of equations and variables grows even faster. To the best of our knowledge, only a handful of examples of $F$-symbols are known where the category has multiplicity [4–7].

The problem of solving the pentagon equations is further complicated by gauge freedom in the solution. When we refer to *a* solution of the pentagon equations, we are really referring to an equivalence class of solutions related by gauge transformations. In the multiplicity free case, a typical approach to finding a set of $F$-symbols begins by determining which $F$-symbols are necessarily zero. In this case, gauge freedom is simply a scale, so it can be used to fix many of the $F$-symbols. When there is multiplicity, the gauge freedom corresponds to basis transformations on nontrivial vector spaces, so gauge fixing is far more intricate.

Due to the challenge in obtaining a set of $F$-symbols, it is valuable to make full use of any solutions that can be obtained. In this work, we exploit the Morita equivalence class of some fusion category $\mathcal{C}$ whose data are known, in order to obtain the $F$-symbols of other categories in the class. In particular, one can use the fact that the category $\mathcal{C}^*_{\mathcal{M}}$ of endomorphisms of some module category $\mathcal{M}$ (over $\mathcal{C}$) yields another category in the Morita equivalence class. We show how tube category techniques can be used to extract the data of this category, expanding on the example in Ref. [8].

The advantage of this method is that we never have to solve the pentagon equations of the complicated category. As input, we can choose the simplest category in the Morita equivalence class (or any category in the equivalence class whose $F$-symbols are already known) and only need to solve the pentagon equations for the module category. As these equations are only of degree two, in contrast to degree three of the pentagon equations in the original category, they are generally easier to solve. Furthermore, since the $F$-symbols from the input category are already gauge fixed, the associators in the module category have less gauge freedom.

This method can be applied to any Morita equivalence class for which the data for a single category, and a module, is known. As an illustration of the power, and use-case, of this technique, we apply it to the Morita equivalence class of fusion categories coming from the Haagerup subfactor [9]. This class consists of three fusion categories, $\mathcal{H}_1$, $\mathcal{H}_2$,

* [danielbarter@gmail.com](danielbarter@gmail.com); [danielbarter.github.io](danielbarter.github.io); [ORCID:0000-0002-6423-117X](ORCID:0000-0002-6423-117X)
† [jcbridgeman1@gmail.com](jcbridgeman1@gmail.com); [jcbridgeman.bitbucket.io](jcbridgeman.bitbucket.io); [ORCID:0000-0002-5638-6681](ORCID:0000-0002-5638-6681)
‡ [rawolf@phys.ethz.ch](rawolf@phys.ethz.ch); [ramonawolf.com](ramonawolf.com); [ORCID:0000-0002-9404-5781](ORCID:0000-0002-9404-5781)

and $\mathcal{H}_3$. The categories $\mathcal{H}_2$ and $\mathcal{H}_3$ are multiplicity free and their $F$-symbols are known [10–13], while the $F$-symbols for $\mathcal{H}_1$, which has multiplicities, have not yet been computed. This demonstrates the degree to which multiplicities increase the difficulty of solving the pentagon equation: even though $\mathcal{H}_1$ has only rank 4 while $\mathcal{H}_2$ and $\mathcal{H}_3$ have rank 6, its $F$-symbols have not been obtained so far.

This paper is organized as follows: In Section II, we review fusion and module categories, and introduce notation for the remainder of the manuscript. Additionally, we review the module tube category. Finally, we discuss unitary structures on each type of category. In Section III, we introduce the algorithm that takes a fusion category and a module category, and returns the categorical data for a Morita equivalent fusion category. We then illustrate the algorithm for the simple example $\mathbf{Vec}(\mathbb{Z}/2\mathbb{Z}) \curvearrowright \mathbf{Vec}$ in Section IV. In Section V, we discuss the Haagerup fusion categories. We illustrate the $F$-symbols we obtain for the category $\mathcal{H}_1$ using the algorithm discussed in this work. To the best of our knowledge, this is the first time these data have been obtained. We conclude in Section VI.

In Appendix A, we briefly discuss the relationship between module functors and tube algebra representations. We provide two additional worked examples, namely $\mathbf{Vec}(S_3) \curvearrowright \mathbf{Vec}$ in Appendix B, and $\mathbf{Rep}(S_3) \curvearrowright \mathbf{Rep}(S_3)$ in Appendix C. Accompanying this manuscript is a collection of Mathematica notebooks that implement the algorithm described here, and include $F$-symbols for the Haagerup category $\mathcal{H}_1$. The code is available at Ref. [14].

## II.  PRELIMINARIES

**Definition 1** (Skeletal fusion category)**.** We sketch a definition of a skeletal fusion category suitable for our purposes. For a more complete (rigorous) definition, we refer the mathematically inclined to Refs. [3 and 15], and the physically inclined to Refs. [16 and 17].

A skeletal fusion category $\mathcal{C}$ consists of the following data:

- A finite set of simple objects $\mathrm{Irr}(\mathcal{C}) = \{1, a, b, \ldots\}$, where $|\mathrm{Irr}(\mathcal{C})|$ is known as the *rank* of $\mathcal{C}$.

- For each triple of simple objects, non-negative integers $N_{ab}^c$ called *fusion coefficients*, obeying

$$N_{1x}^y = N_{x1}^y = \delta_{x,y} \tag{unit}$$

$$\sum_{e \in \mathrm{Irr}(\mathcal{C})} N_{ab}^e N_{ec}^d = \sum_{f \in \mathrm{Irr}(\mathcal{C})} N_{af}^d N_{bc}^f \tag{associativity}$$

For each $x \in \mathrm{Irr}(\mathcal{C})$, there is a unique $\bar{x} \in \mathrm{Irr}(\mathcal{C})$ such that $N_{xy}^1 = N_{yx}^1 = \delta_{y,\bar{x}}$. (duals)

- For each triple of simple objects, a $\mathbb{C}$-vector space $\mathcal{C}(a \otimes b, c)$, called the *fusion space*, of dimension $N_{ab}^c$.

- Associator isomorphisms $\oplus_e \mathcal{C}(a \otimes b, e) \otimes \mathcal{C}(e \otimes c, d) \cong \oplus_f \mathcal{C}(a \otimes f, d) \otimes \mathcal{C}(b \otimes c, f)$ obeying the *pentagon axiom* (Eq. 2.2 of Ref. [3]).

If any of the fusion coefficients is larger than one, we say $\mathcal{C}$ has multiplicity.

It is convenient to specify a basis for all $\mathcal{C}(a \otimes b, c)$, and use a graphical notation commonly referred to as string diagrams when discussing fusion categories. A basis vector in $\mathcal{C}(a \otimes b, c)$ is indicated by a trivalent vertex

$$\alpha \in \mathcal{C}(a \otimes b, c) \leftrightarrow \quad \raisebox{-1em}{\text{$\alpha$}}\ \vcenter{\hbox{$\overset{c}{\underset{a\quad b}{\Large\curlywedge}}$}}\ , \tag{1}$$

while more general vectors correspond to weighted sums of such vertices. Tensor products of vectors are indicated using more complex diagrams, for example

$$\alpha \otimes \beta \in \mathcal{C}(a \otimes b, e) \otimes \mathcal{C}(e \otimes c, d) \leftrightarrow \quad \vcenter{\hbox{$\overset{d}{\underset{a\quad b\quad c}{\Large\text{tree}}}$}}\ . \tag{2}$$

With bases fixed, the associator isomorphisms are realized by a collection of invertible matrices

$$
\begin{array}{c} \text{(diagram)} \end{array} = \sum_{f,\mu,\nu} \left[ F^d_{abc} \right]_{(\alpha,e,\beta)(\mu,f,\nu)} \begin{array}{c} \text{(diagram)} \end{array} , \tag{3}
$$

called the $F$-symbols. We adopt the convention that objects in $\mathrm{Irr}(\mathcal{C})$ are labeled by Roman letters, and basis vectors by Greek letters. Correspondingly, sums over objects run over $\mathrm{Irr}(\mathcal{C})$, while sums over Greek indices run over a complete basis of the appropriate vector space. In this framework, the pentagon equation constraining the $F$-symbols is

$$
\sum_{\zeta} \left[ F^e_{fcd} \right]_{(\beta,g,\gamma)(\rho,x,\zeta)} \left[ F^e_{abx} \right]_{(\alpha,f,\zeta)(\sigma,y,\tau)} = \sum_{z,\lambda,\mu,\nu} \left[ F^g_{abc} \right]_{(\alpha,f,\beta)(\lambda,z,\mu)} \left[ F^e_{azd} \right]_{(\mu,g,\gamma)(\nu,y,\tau)} \left[ F^y_{bcd} \right]_{(\lambda,z,\nu)(\rho,x,\sigma)}. \tag{4}
$$

Changing basis on the $\mathcal{C}(a \otimes b, c)$ spaces leads to a gauge redundancy in the $F$-symbols, meaning that $F$ and $G$ describe the same category, where

$$
\begin{array}{c} \text{(diagram)} \end{array} = \sum_{\beta} [M^c_{ab}]_{\alpha\beta} \begin{array}{c} \text{(diagram)} \end{array} \tag{5a}
$$

$$
\left[ G^d_{abc} \right]_{(\alpha,e,\beta)(\mu,f,\nu)} = \sum_{\gamma,\delta,\sigma,\tau} \left[ F^d_{abc} \right]_{(\gamma,e,\delta)(\sigma,f,\tau)} \left[ (M^e_{ab})^{-1} \right]_{\alpha\gamma} \left[ M^d_{af} \right]_{\tau\nu} \left[ M^f_{bc} \right]_{\sigma\mu} \left[ (M^d_{ec})^{-1} \right]_{\beta\delta} \tag{5b}
$$

where $M$ is an invertible change-of-basis and $\bullet$ indicates the new trivalent basis.

Partial gauge fixing can be used to ensure that

$$
\left[ F^d_{1bc} \right]_{(1,b,\beta)(\mu,d,1)} = \delta_{\beta,\mu}, \qquad \left[ F^d_{a1c} \right]_{(1,a,\beta)(1,c,\nu)} = \delta_{\beta,\nu}, \qquad \left[ F^d_{ab1} \right]_{(\alpha,d,1)(1,b,\nu)} = \delta_{\alpha,\nu}. \tag{6}
$$

For simplicity, we assume such a gauge is chosen for all following computations.

Unitary case

A particularly important class of fusion categories are called unitary. By choosing the bases appropriately, the $F$-symbols of such a category can be transformed into unitary matrices. In the unitary case, we can additionally fix the gauge to ensure that

$$
\left[ F^a_{a\bar{a}a} \right]_{(1,1,1)(1,1,1)} = \frac{\varkappa_a}{d_a}, \tag{7}
$$

where $\varkappa_a = \pm 1$, and $d_a$ is the Frobenius-Perron dimension of $a$ completely defined by

$$
d_a > 0 \tag{8a}
$$

$$
d_a d_b = \sum_c N^c_{ab} d_c. \tag{8b}
$$

A covector in the dual space to $\mathcal{C}(a \otimes b, c)$ is indicated via a 'splitting' vertex, with basis defined by

$$
\alpha \in \mathcal{C}(c, a \otimes b) \leftrightarrow \begin{array}{c} \text{(diagram)} \end{array} , \qquad \begin{array}{c} \text{(diagram)} \end{array} = \sqrt{\frac{d_a d_b}{d_c}} \delta_{\alpha,\beta} \delta_{c,e} \begin{array}{c} \text{(diagram)} \end{array} . \tag{9}
$$

Re-association of covectors is also given by the $F$-symbols

$$
\begin{gathered}
\vcenter{\hbox{
\begin{tikzpicture}
\end{tikzpicture}
}}
\end{gathered}
= \sum_{f,\mu,\nu} \overline{\left[F^d_{abc}\right]}_{(\alpha,e,\beta)(\mu,f,\nu)}
\;\;\;\;
\vcenter{\hbox{
\begin{tikzpicture}
\end{tikzpicture}
}}
\;\;,
\tag{10}
$$

where $\bar{\cdot}$ is the complex conjugate.

**Definition 2** ($\mathcal{C}$-module category). We sketch a definition of a skeletal left $\mathcal{C}$-module category suitable for our purposes. For a more complete (rigorous) definition, we refer to Ref. [3].

Given a skeletal fusion category $\mathcal{C}$, with specified bases for all fusion spaces, a skeletal $\mathcal{C}$-module category $\mathcal{M}$ consists of the following data:

- A finite set of simple objects $\mathrm{Irr}(\mathcal{M}) = \{m, n, \ldots\}$.

- For each pair of simple objects $m, n \in \mathrm{Irr}(\mathcal{M})$, and simple object $a \in \mathrm{Irr}(\mathcal{C})$, non-negative integers $N^n_{am}$ called *fusion coefficients*, obeying

$$
\begin{aligned}
N^n_{1m} &= \delta_{m,n} &&\text{(unit)} \\
\sum_{e \in \mathrm{Irr}(\mathcal{C})} N^e_{ab} N^n_{em} &= \sum_{p \in \mathrm{Irr}(\mathcal{M})} N^n_{ap} N^p_{bm} &&\text{(associativity)}
\end{aligned}
$$

- For each pair of simple objects $m, n \in \mathrm{Irr}(\mathcal{M})$, and simple object $a \in \mathrm{Irr}(\mathcal{C})$, a $\mathbb{C}$-vector space $\mathcal{M}(a \triangleright m, n)$ of dimension $N^n_{am}$.

- Associator isomorphisms $\oplus_e \mathcal{C}(a \otimes b, e) \otimes \mathcal{M}(e \triangleright m, n) \cong \oplus_p \mathcal{M}(a \triangleright p, n) \otimes \mathcal{M}(b \triangleright m, p)$ obeying the *module pentagon axiom* (Eq. 7.2 of Ref. [3]).

If any of the fusion coefficients is larger than one, we say $\mathcal{M}$ has multiplicity.

Again, it is convenient to specify bases for all $\mathcal{M}(a \triangleright m, n)$, and extend the string diagram notation. A basis vector in $\mathcal{M}(a \triangleright m, n)$ is indicated by a trivalent vertex

$$
\alpha \in \mathcal{M}(a \triangleright m, n) \leftrightarrow
\vcenter{\hbox{
\begin{tikzpicture}
\end{tikzpicture}
}}
\;\;,
\tag{11}
$$

while more general vectors correspond to weighted sums of such vertices. Tensor products of vectors are indicated using more complex diagrams, for example

$$
\alpha \otimes \beta \in \mathcal{C}(a \otimes b, e) \otimes \mathcal{M}(e \triangleright m, n) \leftrightarrow
\vcenter{\hbox{
\begin{tikzpicture}
\end{tikzpicture}
}}
\;\;.
\tag{12}
$$

With bases fixed, the associator isomorphisms are realized by a collection of invertible matrices

$$
\vcenter{\hbox{
\begin{tikzpicture}
\end{tikzpicture}
}}
= \sum_{p,\mu,\nu} \left[L^n_{abm}\right]_{(\alpha,e,\beta)(\mu,p,\nu)}
\vcenter{\hbox{
\begin{tikzpicture}
\end{tikzpicture}
}}
\;\;,
\tag{13}
$$

called the $L$-symbols.

In this framework, the mixed pentagon equation constraining the $L$-symbols is

$$
\sum_{\zeta} \left[L^n_{fcm}\right]_{(\beta,g,\gamma)(\rho,p,\zeta)} \left[L^n_{abp}\right]_{(\alpha,f,\zeta)(\sigma,q,\tau)} = \sum_{z,\lambda,\mu,\nu} \left[F^g_{abc}\right]_{(\alpha,f,\beta)(\lambda,z,\mu)} \left[L^n_{azm}\right]_{(\mu,g,\gamma)(\nu,q,\tau)} \left[L^q_{bcm}\right]_{(\lambda,z,\nu)(\rho,p,\sigma)},
\tag{14}
$$

where the $F$-symbol is that of the underlying fusion category $\mathcal{C}$.

Changing basis on the $\mathcal{M}(a \triangleright m, n)$ spaces (holding the bases in $\mathcal{C}$ fixed) leads to a gauge redundancy in the $L$-symbols, meaning that $L$ and $\tilde{L}$ describe the same category, where

$$
\begin{array}{c} n \\ \alpha \\ a\, m \end{array} = \sum_{\beta} [M_{am}^n]_{\alpha\beta}\ \begin{array}{c} n \\ \beta \\ a\, m \end{array} \tag{15a}
$$

$$
\left[\tilde{L}_{abm}^n\right]_{(\alpha,e,\beta)(\mu,p,\nu)} = \sum_{\delta,\sigma,\tau} \left[L_{abm}^n\right]_{(\alpha,e,\delta)(\sigma,p,\tau)} \left[M_{ap}^n\right]_{\tau\nu} [M_{bm}^p]_{\sigma\mu} \left[(M_{em}^n)^{-1}\right]_{\beta\delta}. \tag{15b}
$$

Partial gauge fixing can be used to ensure that

$$
\left[L_{1bm}^n\right]_{(1,b,\beta)(\mu,n,1)} = \delta_{\beta,\mu}, \qquad\qquad \left[L_{a1m}^n\right]_{(1,a,\beta)(1,m,\nu)} = \delta_{\beta,\nu}. \tag{16}
$$

For simplicity, we assume such a gauge is chosen for all following computations.

Unitary case

If there is a basis in which the $L$-symbol is unitary as a matrix[1], the module category is called *unitary*.

A covector in the dual space to $\mathcal{C}(a \triangleright m, n)$ is indicated via a 'splitting' vertex, with basis defined by

$$
\alpha \in \mathcal{M}(n, a \triangleright m) \leftrightarrow \begin{array}{c} a\, m \\ \alpha \\ n \end{array}, \qquad\qquad a\left(\begin{array}{c} n \\ \beta \\ p \\ \alpha \\ m \end{array}\right) = \sqrt{\tfrac{d_a d_p}{d_n}}\,\delta_{\alpha,\beta}\delta_{m,n}\ \begin{array}{c} n \\ \big| \\ m \end{array}, \tag{17}
$$

where $d_m$ is the *Frobenius-Perron dimension* of $m$ completely defined by

$$
d_m > 0 \tag{18a}
$$

$$
d_a d_m = \sum_{n \in \mathrm{Irr}(\mathcal{M})} N_{am}^n d_n \tag{18b}
$$

$$
\sum_{m \in \mathrm{Irr}(\mathcal{M})} d_m^2 = \sum_{a \in \mathrm{Irr}(\mathcal{C})} d_a^2. \tag{18c}
$$

Re-association of covectors is also given by the $L$-symbols

$$
\begin{array}{c} a \quad b \quad m \\ \alpha \\ e \quad \beta \\ n \end{array} = \sum_{p,\mu,\nu} \overline{\left[L_{abm}^n\right]}_{(\alpha,e,\beta)(\mu,p,\nu)} \begin{array}{c} a \quad b \quad m \\ \mu \\ p \\ \nu \\ n \end{array}, \tag{19}
$$

where $\bar{\ }$ is the complex conjugate.

For all following discussions, we assume that $\mathcal{M}$ is indecomposable as a $\mathcal{C}$-module category, meaning $\mathcal{M}$ cannot be decomposed as a direct sum of module categories. If we do not restrict $\mathcal{M}$ in this way, the result of the algorithm we present will be *multifusion*. We refer to Ref. [3] for more details.

---

[1] We also require this to be compatible with the pivotal/unitary structure on $\mathcal{C}$.

**Definition 3** (Module tube category). Given a fusion category $\mathcal{C}$, and a $\mathcal{C}$-module category $\mathcal{M}$, the module tube category $\mathbf{Tub}_{\mathcal{C}}(\mathcal{M})$ has as objects pairs

$$\mathrm{Ob}(\mathbf{Tub}_{\mathcal{C}}(\mathcal{M})) = \{(m, n) \, | \, m, n \in \mathrm{Ob}(\mathcal{M})\}. \tag{20}$$

Given a pair of simple objects $(m, n)$, $(p, q)$, a basis for the morphism space $\mathbf{Tub}_{\mathcal{C}}(\mathcal{M})((m, n), (p, q))$ is given by the set of diagrams

$$\Lambda := \left\{ \mathbf{T}[mn|pq]_{\alpha, x, \beta} := \vcenter{\hbox{}} \;\middle|\; x \in \mathrm{Irr}(\mathcal{C}), 1 \le \alpha \le N^{p}_{xm}, 1 \le \beta \le N^{q}_{xn} \right\}. \tag{21}$$

Composition of morphisms is evaluated using the re-association matrices $F$ and $L$ from the underlying categories

$$\mathbf{T}[m'n'|p'q']_{\alpha', x', \beta'} \circ \mathbf{T}[mn|pq]_{\alpha, x, \beta} = \vcenter{\hbox{}} \tag{22a}$$

$$= \delta_{p,m'} \delta_{q,n'} \sum_{y, \zeta, \sigma, \tau} \sqrt{\frac{d_x d_{x'}}{d_y}} \left[ \left( L^{q'}_{x'xn} \right)^{-1} \right]_{(\beta, q, \beta')(\zeta, y, \tau)} \left[ \left( \tilde{L}^{p'}_{x'xm} \right)^{-1} \right]_{(\alpha, p, \alpha')(\zeta, y, \sigma)} \mathbf{T}[mn|p'q']_{\sigma, y, \tau}, \tag{22b}$$

Where $\tilde{L}$ is the $L$-symbol for splitting vertices.

With this composition, the set of all morphisms forms an algebra closely related to Ocneanu's *tube algebra* [18]. Since it will not cause confusion in the current context, we will refer to the algebra Eq. (21) as the tube algebra. This algebra is associative due to the pentagon equation Eq. (14), and unital. When the module $\mathcal{M}$ is irreducible, the tube algebra is semisimple [3, 19], and therefore isomorphic to a direct sum of $\mathbb{C}$-matrix algebras. This becomes important when computing representations of this algebra in the following sections.

Finally, we define a tensor product via diagrams

$$\mathbf{T}[mn|pq]_{\alpha, x, \beta} \otimes \mathbf{T}[m'n'|p'q']_{\alpha', x', \beta'} := \delta_{q, p'} \vcenter{\hbox{}} \;, \tag{23}$$

which is acted on by tube diagrams $\mathbf{T}[p'q'|rs]_{\sigma, y, \tau}$ by acting on the 'outside' and reducing using the string manipulation rules.

Given a fusion category $\mathcal{C}$, and a finite, irreducible module category $\mathcal{C} \curvearrowright \mathcal{M}$, we denote the category of $\mathcal{C}$-module endofunctors from $\mathcal{M}$ to itself by $\mathcal{C}^{*}_{\mathcal{M}}$ [20–22]. This category has a natural tensor structure, given by functor composition. In Appendix A, we briefly recall this structure, and how tube algebra representations relate to these endofunctors.

**Definition 4** (Morita equivalence). Let $\mathcal{C}$, $\mathcal{D}$ be fusion categories. We say that $\mathcal{C}$ and $\mathcal{D}$ are *Morita equivalent* if there exists an irreducible $\mathcal{C}$-module $\mathcal{M}$ such that $\mathcal{C}^{*}_{\mathcal{M}}$ is equivalent to $\mathcal{D}$. Notice that in this case, $\mathcal{M}$ is an irreducible $\mathcal{C}$–$\mathcal{D}$ module.

*Unitary case*

When $\mathcal{C}$ and $\mathcal{M}$ are unitary, the tube algebra comes equipped with an induced $*$-structure, which exchanges the inner and outer (source and target) circles in the diagram

$$\mathbf{T}[mn|pq]_{\alpha,x,\beta}^* := \bar{x} \left( \begin{array}{c} n \\ \beta \\ q \\ p \\ \alpha \\ m \end{array} \right) = \sum_{\sigma,\tau} d_x \sqrt{\frac{d_m d_n}{d_p d_q}} \overline{\left[L_{\bar{x}xm}^m\right]}_{(1,1,1)(\alpha,p,\sigma)} \left[L_{\bar{x}xn}^n\right]_{(1,1,1)(\beta,q,\tau)} \mathbf{T}[pq|mn]_{\sigma,\bar{x},\tau}. \tag{24}$$

Additionally, the algebra is equipped with a linear functional and associated inner product

$$\omega(\mathbf{T}[mn|pq]_{\alpha,x,\beta}) = \delta_{x,1} d_m d_n \tag{25a}$$

$$\left\langle \mathbf{T}[m'n'|p'q']_{\alpha',x',\beta'}, \mathbf{T}[mn|pq]_{\alpha,x,\beta} \right\rangle := \omega(\mathbf{T}[m'n'|p'q']_{\alpha',x',\beta'}^* \circ \mathbf{T}[mn|pq]_{\alpha,x,\beta}). \tag{25b}$$

The form $\langle \cdot, \cdot \rangle$ is linear in the second argument by definition. It can readily (although tediously) be verified that $\langle A, B \rangle = \overline{\langle B, A \rangle}$. Showing that $\langle A, A \rangle > 0 \, \forall A \neq 0$ reduces to showing that

$$\left[L_{\bar{x}xa}^a\right]_{(1,1,1)(\alpha,b,\sigma)} = 0 \, \forall \sigma \implies b \notin x \triangleright a. \tag{26}$$

This follows from bending

$$\varkappa_x \quad \begin{array}{c} b \\ \alpha \\ x \quad a \end{array} = \begin{array}{c} b \\ \bar{x} \quad \alpha \\ x \quad a \end{array} = d_x \sum_{\sigma,\tau} \overline{\left[L_{\bar{x}xa}^a\right]}_{(1,1,1)(\alpha,b,\sigma)} \left[L_{x\bar{x}b}^b\right]_{(1,1,1)(\sigma,a,\tau)} \begin{array}{c} b \\ \tau \\ x \quad a \end{array} \tag{27}$$

It is necessary to define a balanced inner product on the tensor product space [23]

$$\left\langle \mathbf{T}[rs|tu]_{\sigma,y,\tau} \otimes \mathbf{T}[r's'|t'u']_{\sigma',y',\tau'}, \mathbf{T}[mn|pq]_{\alpha,x,\beta} \otimes \mathbf{T}[m'n'|p'q']_{\alpha',x',\beta'} \right\rangle :=$$

$$\frac{\left\langle \mathbf{T}[rs|tu]_{\sigma,y,\tau}, \mathbf{T}[mn|pq]_{\alpha,x,\beta} \right\rangle \left\langle \mathbf{T}[r's'|t'u']_{\sigma',y',\tau'}, \mathbf{T}[m'n'|p'q']_{\alpha',x',\beta'} \right\rangle}{\sqrt{d_q d_u}}. \tag{28}$$

If $\mathcal{C}$ is a unitary fusion category and trivalent vertices are chosen to be compatible with the unitary structure on $\mathcal{C}$, then the $F$-symbol is a unitary matrix. If $\mathcal{M}$ is a unitary $\mathcal{C}$-module, then $\mathcal{C}_\mathcal{M}^*$ is a unitary fusion category [24]. In this paper, we demonstrate how to compute a basis set of trivalent vertices for $\mathcal{C}_\mathcal{M}^*$ compatible with its unitary structure. In particular, this implies that the computed associator for $\mathcal{C}_\mathcal{M}^*$ is unitary. The procedure goes as follows:

1. $\mathbf{Tub}_\mathcal{C}(\mathcal{M})$ inherits a $*$-structure and trace from the unitary structures on $\mathcal{C}$ and $\mathcal{M}$. Compute matrix units $e_{ij}$ in $\mathbf{Tub}_\mathcal{C}(\mathcal{M})$ which satisfy $e_{ij}^* = e_{ji}$. Then $\mathbf{Tub}_\mathcal{C}(\mathcal{M})e_{ii}$ is an irreducible unitary representation of $\mathbf{Tub}_\mathcal{C}(\mathcal{M})$, its unitary structure inherited from the inclusion $\mathbf{Tub}_\mathcal{C}(\mathcal{M})e_{ii} \hookrightarrow \mathbf{Tub}_\mathcal{C}(\mathcal{M})$.

2. Given unitary representations $V_i, V_j, V_k$ of $\mathbf{Tub}_\mathcal{C}(\mathcal{M})$, choose a basis of intertwiners $V_i \otimes V_j \to V_k$ which are isometric projections. With respect to this basis, the $F$-symbol of $\mathbf{Tub}_\mathcal{C}(\mathcal{M})$ is unitary.

## III.  COMPUTING DATA FOR $\mathcal{C}_\mathcal{M}^*$

Given a fusion category $\mathcal{C}$, and an indecomposable $\mathcal{C}$-module category $\mathcal{M}$, the first piece of data defining $\mathcal{C}_\mathcal{M}^*$ is the set of simple objects. We compute this by constructing a complete set of irreducible representations of the tube algebra $\mathbf{Tub}_\mathcal{C}(\mathcal{M})$. Specifically, since $\mathbf{Tub}_\mathcal{C}(\mathcal{M})$ is semisimple, we can compute an explicit *Artin-Wedderburn* isomorphism

$$\mathbf{Tub}_\mathcal{C}(\mathcal{M}) \cong \oplus_{\alpha=1}^n \text{Mat}(D_\alpha), \tag{29}$$

where $\mathrm{Mat}(D)$ is the $D \times D$ matrix algebra over $\mathbb{C}$, and $\sum_{\alpha=1}^{n} D_{\alpha}^2 = \dim \mathbf{Tub}_{\mathcal{C}}(\mathcal{M})$. In particular, it is convenient to fix a *matrix unit* basis for $\mathrm{Mat}(D_\alpha)$,

$$\left\{ [e_\alpha]_{ij} \,\Big|\, 0 \le i,j < D_\alpha, [e_\alpha]_{ij}[e_\alpha]_{kl} = \delta_{j,k}[e_\alpha]_{il} \right\}, \tag{30}$$

and seek a solution to the set of equations

$$[e_\alpha]_{ij} = \sum_{P \in \Lambda} C_P^{(\alpha)} P, \tag{31}$$

where $\Lambda$ is the basis defined in Eq. (21), and $C_P^{(\alpha)}$ are coefficients to be determined. Although it is in principle computationally hard to find such an isomorphism, in practice it can be solved (or accurately approximated) in many cases. This is discussed in Ref. [8], and example code is provided there.

Given such an isomorphism, we can construct a vector space with basis

$$[v_\alpha]_i := [e_\alpha]_{i0}, \tag{32}$$

forming an irreducible representation (irrep) of $\mathbf{Tub}_{\mathcal{C}}(\mathcal{M})$. In Refs. [8, 25, and 26], these vector spaces were called binary interface defects, since physically they correspond to excitations at the interface of two boundaries. It is convenient to extend our graphical notation to include these vectors

$$[v_\alpha]_i = \quad \alpha \bullet i \quad , \tag{33}$$

where the left label denotes the irrep label, and the right index specifies a basis vector in that representation. Omission of the vector label indicates the full representation. Each irreducible representation is a simple object in the category $\mathcal{C}_{\mathcal{M}}^*$. Fusion of objects corresponds to tensoring representations. Using this notation, the tensor product of two representations is indicated by vertical stacking

$$\alpha \otimes \beta := \quad \begin{matrix} \beta \bullet \\ \alpha \bullet \end{matrix} \quad . \tag{34}$$

The tensor product space comes equipped with an action of $\mathbf{Tub}_{\mathcal{C}}(\mathcal{M})$, and so forms a (potentially reducible) representation. Decomposing into irreps gives the fusion rules of the fusion category $\mathcal{C}_{\mathcal{M}}^*$

$$\begin{matrix} \beta \bullet \\ \alpha \bullet \end{matrix} \cong \bigoplus_{\gamma} \quad \gamma \bullet \quad , \tag{35}$$

where $\gamma$ runs over the irreps occurring (possibly multiple times) in the decomposition of $\alpha \otimes \beta$. The fusion rules $N_{\alpha\beta}^{\gamma}$ can be deduced by projecting generic vectors in the tensor product

$$v = \sum_{i,j} C_{ij} \quad \begin{matrix} \beta \bullet j \\ \alpha \bullet i \end{matrix} \tag{36}$$

onto the irrep $\gamma$ using the identity matrix

$$\mathbb{1}_\gamma = \sum_i [e_\gamma]_{ii}. \tag{37}$$

The fusion $N_{\alpha\beta}^{\gamma}$ is the dimension of the space spanned by such projected vectors.

Explicit trivalent vertices for the category $\mathcal{C}_{\mathcal{M}}^*$ can be computed by forming matrices for the isomorphisms Eq. (35). Since there was a great deal of freedom in the choice of basis $[v_\alpha]_i$, these matrices are far from unique. We can change basis on all three of the involved tube algebra representations. Choosing distinct bases will lead to distinct, but equivalent, $F$-symbols.

We denote a map embedding the irrep $\gamma$ into the representation $\alpha \otimes \beta$ by

$$V_{\alpha\beta}^{\gamma;x} := \quad \begin{matrix} \beta \\ \alpha \end{matrix} \!\!>\!\!\!\overset{x}{\phantom{x}}\!\!\!\!- \gamma \quad , \tag{38}$$

a matrix with $\dim \alpha \otimes \beta$ rows and $\dim \gamma$ columns. It is convenient to reshape this matrix into a 3-tensor, however $\dim \alpha \otimes \beta$ may not be a composite number due to the tensor product rule Eq. (23) requiring matching of the middle strand label. For this reason, it is useful to fill out with zero rows, corresponding to cases where $[v_\alpha]_i \otimes [v_\beta]_j = 0$. Following this process, the matrix can be recast as a 3-tensor of size $(\dim \alpha, \dim \beta; \dim \gamma)$. If there are multiple copies of $\gamma \in \alpha \otimes \beta$, there will be multiple such matrices, forming a vector space. We choose a basis of matrices, and label the vertex to identify which basis vector is being referred to. The choice here corresponds to the gauge freedom in choosing a basis for the fusion space.

Assuming $N_{\alpha\beta}^\gamma \neq 0$, matrix elements for $V_{\alpha\beta}^\gamma$ can be computed as follows:

- Pick a generic vector $v \in \alpha \otimes \beta$ (Eq. (36)).

- Project onto the target irrep $\gamma$ using $[e_\gamma]_{00}$.

  If $N_{\alpha\beta}^\gamma = 1$, call the result $[v_\gamma]_0$ ($[v_\gamma]_0$ unique up to scale).

  If $N_{\alpha\beta}^\gamma > 1$, repeat until $N_{\alpha\beta}^\gamma$ independent vectors $[v_{\gamma,x}]_0$ are obtained. If an inner product is defined, these could be made orthonormal.

- Build the rest of the basis $[v_{\gamma,x}]_i = [e_\gamma]_{i0}[v_{\gamma,x}]_0$.

- The entries in the $i$th column of $V_{\alpha\beta}^{\gamma;x}$ are the coefficients of $[v_{\gamma,x}]_i$ in the tensor product basis.

Given these 3-tensors, there are two ways they can be combined into intertwining maps $\delta \to \alpha \otimes \beta \otimes \delta$. These provide two bases for the intertwiner space, and are related by a change of basis matrix,

$$
\begin{array}{c}
\gamma \\
\beta \quad \diagdown \, j \\
\quad \diagup\!\!\diagdown_\mu \, \delta \\
\alpha \diagup i
\end{array}
\;=\; \sum_{k,\nu,l} \left[ F_{\alpha\beta\gamma}^\delta \right]_{(i,\mu,j)(k,\nu,l)}
\begin{array}{c}
\gamma \diagdown k \\
\quad \diagdown_\nu \\
\beta \diagup\!\!\diagdown \, \delta \\
\quad l \\
\alpha \diagup
\end{array}
. \tag{39}
$$

Solving this (linear) equation gives $F$-matrices for $\mathcal{C}_\mathcal{M}^*$, which, in general, are distinct from those of the input category $\mathcal{C}$.

To summarize the algorithm:

1. Compute irreducible representations of tube algebra.

2. Compute decomposition of tensor product of all irrep pairs.

3. Form explicit matrices for isomorphism, express as 3-tensors.

4. Solve linear equations Eq. (39) to obtain (new) $F$-symbols.

*Unitary case*

When $\mathcal{C}$ and $\mathcal{M}$ are unitary, it is useful to respect the $*$-structure when solving Eq. (31). In particular, we should solve Eq. (31) with the additional condition that

$$
\left( [e_\alpha]_{ij} \right)^* = [e_\alpha]_{ji}. \tag{40}
$$

This ensures that our resulting tube representations are unitary (although, our computed bases are not necessarily orthonormal). Since the tube representations are unitary, we can insist that the embedding maps Eq. (38) are isometric with respect to the equipped norms, and distinct maps $V_{\alpha\beta}^{\gamma;x}, V_{\alpha\beta}^{\gamma;y}$ obey

$$
\left\langle V_{\alpha\beta}^{\gamma;x}([v_\gamma]_i), V_{\alpha\beta}^{\gamma;y}([v_\gamma]_j) \right\rangle = \delta_{x,y} \left\langle [v_\gamma]_i, [v_\gamma]_j \right\rangle. \tag{41}
$$

As outlined in the preliminaries, it makes sense that choosing vertices in this way leads to a unitary gauge for the resulting $F$-symbols, and we have numerically verified this for both of the Haagerup categories considered in this

paper. It can be verified generally as follows

$$\delta_{(w,\mu,x),(y,\nu,z)}\left\langle [v_\delta]_i, [v_\delta]_j \right\rangle = \left\langle \begin{array}{c} \gamma \\ \beta \\ \alpha \end{array} \begin{array}{c} x \\ i \\ \mu \\ w \end{array} \delta \ , \ \begin{array}{c} \gamma \\ \beta \\ \alpha \end{array} \begin{array}{c} z \\ j \\ \nu \\ y \end{array} \delta \ , \right\rangle \tag{42a}$$

$$= \sum_{a,\epsilon,b,c,\sigma,d} \overline{\left[ F^\delta_{\alpha\beta\gamma} \right]}_{(w,\mu,x)(a,\epsilon,b)} \left[ F^\delta_{\alpha\beta\gamma} \right]_{(y,\nu,z)(c,\sigma,d)} \left\langle \begin{array}{c} \gamma \\ \beta \\ \alpha \end{array} \begin{array}{c} a \\ \epsilon \\ i \\ b \end{array} \delta \ , \ \begin{array}{c} \gamma \\ \beta \\ \alpha \end{array} \begin{array}{c} c \\ \sigma \\ j \\ d \end{array} \delta \right\rangle \tag{42b}$$

$$= \sum_{a,\epsilon,b} \overline{\left[ F^\delta_{\alpha\beta\gamma} \right]}_{(w,\mu,x)(a,\epsilon,b)} \left[ F^\delta_{\alpha\beta\gamma} \right]_{(y,\nu,z)(a,\epsilon,b)} \left\langle [v_\delta]_i, [v_\delta]_j \right\rangle \tag{42c}$$

$$\implies \sum_{a,\epsilon,b} \overline{\left[ F^\delta_{\alpha\beta\gamma} \right]}_{(w,\mu,x)(a,\epsilon,b)} \left[ F^\delta_{\alpha\beta\gamma} \right]_{(y,\nu,z)(a,\epsilon,b)} = \delta_{(w,\mu,x),(y,\nu,z)}. \tag{42d}$$

Alternatively, one could attempt to change the gauge after finding the $F$-symbols, however this is challenging if $\mathcal{C}^*_\mathcal{M}$ has multiplicity.

## IV. A WORKED EXAMPLE: $\mathbf{Vec}(\mathbb{Z}/2\mathbb{Z})^*_{\mathbf{Vec}}$

We work through a particularly simple example to recover the $F$-symbols of $\mathbf{Rep}(\mathbb{Z}/2\mathbb{Z})$ from a module, namely $\mathbf{Vec}$, over $\mathbf{Vec}(\mathbb{Z}/2\mathbb{Z})$.

The skeletal fusion category $\mathbf{Vec}(\mathbb{Z}/2\mathbb{Z})$ has two simple objects, $\{0,1\}$, and fusion rules $a \otimes b := a + b \mod 2$. Both objects have $d_x = 1$. All $F$-symbols are 1 when permitted by fusion. In all cases, we neglect to draw the strings corresponding to the unit object 0. We consider a module category $\mathbf{Vec}$ with a single simple object, denoted $*$ or a blue string, with dimension $d_* = \sqrt{2}$. All module $L-$symbols are 1 when permitted by fusion.

A basis for the tube algebra is given by

$$\Lambda := \left\{ \ \mathbf{T}_0 = \begin{array}{c} \end{array} , \mathbf{T}_1 = \begin{array}{c} \end{array} \right\}. \tag{43}$$

Since all the $F$- and $L$-symbols are 1, the product Eq. (22) reduces to $\mathbf{T}_x \circ \mathbf{T}_y = \mathbf{T}_{x+y \mod 2}$, recovering the group algebra $\mathbb{C}[\mathbb{Z}/2\mathbb{Z}]$. Finally, these categories are equipped with a $*$-structure, which acts trivially on the basis $\Lambda$.

### Step 1.

The tube algebra $\mathbb{C}\Lambda$ decomposes as two copies of the 1-dimensional algebra $\mathbb{C}\Lambda \cong \mathbb{C} \oplus \mathbb{C}$. We will label the two irreducible representations by $1, \psi$. A complete set of matrix units is given by

$$[e_1]_{00} = \frac{\mathbf{T}_0 + \mathbf{T}_1}{2} = \frac{1}{2}\left( \begin{array}{c} \end{array} + \begin{array}{c} \end{array} \right) \tag{44a}$$

$$[e_\psi]_{00} = \frac{\mathbf{T}_0 - \mathbf{T}_1}{2} = \frac{1}{2}\left( \begin{array}{c} \end{array} - \begin{array}{c} \end{array} \right). \tag{44b}$$

Since both representations are 1-dimensional, we neglect the matrix indices for the remainder of this section. A basis for the representations is given by

$$[v_1] = [e_1] \qquad\qquad\qquad [v_\psi] = [e_\psi], \tag{45}$$

with action

$$[e_\alpha][v_\beta] = \delta_{\alpha,\beta}[v_\alpha]. \tag{46}$$

Both basis vectors $[v_x]$ have norm 1.

**Step 2.**

The tensor product basis is 4-dimensional,

$$[v_1] \otimes [v_1] = \frac{1}{4}\left( \ \text{[diagram]} + \text{[diagram]} + \text{[diagram]} + \text{[diagram]} \ \right) \qquad [v_1] \otimes [v_\psi] = \frac{1}{4}\left( \ \text{[diagram]} - \text{[diagram]} + \text{[diagram]} - \text{[diagram]} \ \right) \tag{47a}$$

$$[v_\psi] \otimes [v_1] = \frac{1}{4}\left( \ \text{[diagram]} + \text{[diagram]} - \text{[diagram]} - \text{[diagram]} \ \right) \qquad [v_\psi] \otimes [v_\psi] = \frac{1}{4}\left( \ \text{[diagram]} - \text{[diagram]} - \text{[diagram]} + \text{[diagram]} \ \right). \tag{47b}$$

To obtain the fusion rules for $\mathbf{Vec}(\mathbb{Z}/2\mathbb{Z})^*_{\mathbf{Vec}}$, we project the tensor product basis above onto the irreps. This is done by left multiplication with the basis for the representations given in Eq. (45). Graphically, left multiplication corresponds to putting the tubes given in Eq. (44a) and Eq. (44b) on the outside of the tubes in the tensor product and reducing the diagrams using the $F$- and $L$-symbols. For example,

$$e_\psi([v_1] \otimes [v_\psi]) = \frac{1}{2}\left( \ \text{[diagram]} - \text{[diagram]} \ \right) \circ \frac{1}{4}\left( \ \text{[diagram]} - \text{[diagram]} + \text{[diagram]} - \text{[diagram]} \ \right) \tag{48a}$$

$$= \frac{1}{8}\left( \ \text{[diagram]} - \text{[diagram]} + \text{[diagram]} - \text{[diagram]} - \text{[diagram]} + \text{[diagram]} - \text{[diagram]} + \text{[diagram]} \ \right) \tag{48b}$$

$$= \frac{1}{8}\left( \ \text{[diagram]} - \text{[diagram]} + \text{[diagram]} - \text{[diagram]} - \text{[diagram]} + \text{[diagram]} - \text{[diagram]} + \text{[diagram]} \ \right) = [v_1] \otimes [v_\psi], \tag{48c}$$

which tells us that $\psi$ is in the decomposition of the tensor product $1 \otimes \psi$.

More generally, to compute the multiplicity of an irreducible inside some representation, you compute the dimension of the image of multiplication by the corresponding indecomposable idempotent. Summing up, the fusion rules for $\mathbf{Vec}(\mathbb{Z}/2\mathbb{Z})^*_{\mathbf{Vec}}$ are

$$1 \otimes x = x = x \otimes 1 \qquad\qquad \psi \otimes \psi = 1, \tag{49}$$

where $x \in \{1, \psi\}$.

**Step 3.**

Next, we provide explicit trivalent intertwiners. As discussed in Section III, we ensure that these are isometric. All basis vectors in the tensor product basis $[v_x] \otimes [v_y]$ have norm $2^{-1/4}$, arising from the dimension of the module object $d_* = \sqrt{2}$, and Eq. (28). Recall that these are obtained (for a given choice of irreps to tensor) by: first choosing a generic vector in the tensor product, followed by projecting onto the target irrep. Since all irreps are 1-dimensional in this case, expressing the result as in the tensor product representation completes the computation.

We obtain the isometric intertwiners with matrix representations

$$\begin{matrix} 1 \\ 1 \end{matrix} \!\!\!\!\succ\!\!-\!\! 1 \ = [v_1] \otimes [v_1] \ \begin{pmatrix} [v_1] \\ \omega^1_{11} \end{pmatrix} \times 2^{1/4} \qquad\qquad \begin{matrix} \psi \\ \psi \end{matrix} \!\!\!\!\succ\!\!-\!\! 1 \ = [v_\psi] \otimes [v_\psi] \ \begin{pmatrix} [v_1] \\ \omega^1_{\psi\psi} \end{pmatrix} \times 2^{1/4} \tag{50a}$$

$$\begin{matrix} \psi \\ 1 \end{matrix} \!\!\!\!\succ\!\!-\!\! \psi \ = [v_1] \otimes [v_\psi] \ \begin{pmatrix} [v_\psi] \\ \omega^\psi_{1\psi} \end{pmatrix} \times 2^{1/4} \qquad\qquad \begin{matrix} 1 \\ \psi \end{matrix} \!\!\!\!\succ\!\!-\!\! \psi \ = [v_\psi] \otimes [v_1] \ \begin{pmatrix} [v_\psi] \\ \omega^\psi_{\psi 1} \end{pmatrix} \times 2^{1/4}, \tag{50b}$$

where the $\omega^x_{ab}$'s are complex numbers with $|\omega^c_{ab}| = 1$.

| ⊗ | 1 | α | α² | ρ | αρ | α²ρ |
|---|---|---|---|---|---|---|
| 1 | 1 | α | α² | ρ | αρ | α²ρ |
| α | α | α² | 1 | αρ | α²ρ | ρ |
| α² | α² | 1 | α | α²ρ | ρ | αρ |
| ρ | ρ | α²ρ | αρ | $1+X$ | $α^2+X$ | $α+X$ |
| αρ | αρ | ρ | α²ρ | $α+X$ | $1+X$ | $α^2+X$ |
| α²ρ | α²ρ | αρ | ρ | $α^2+X$ | $α+X$ | $1+X$ |

| $\mathcal{H}_3 \triangleright \mathcal{M}_{3,1}$ | Γ | αΓ | α²Γ | Λ |
|---|---|---|---|---|
| 1 | Γ | αΓ | α²Γ | Λ |
| α | αΓ | α²Γ | Γ | Λ |
| α² | α²Γ | Γ | αΓ | Λ |
| ρ | $Γ+α^2Γ+Λ$ | $Γ+αΓ+Λ$ | $Γ+α^2Γ+Λ$ | Y |
| αρ | $Γ+α^2Γ+Λ$ | $αΓ+α^2Γ+Λ$ | $Γ+αΓ+Λ$ | Y |
| α²ρ | $Γ+αΓ+Λ$ | $Γ+α^2Γ+Λ$ | $αΓ+α^2Γ+Λ$ | Y |

| $\mathcal{H}_3 \triangleright \mathcal{M}_{3,2}$ | G | ρG |
|---|---|---|
| 1 | G | ρG |
| α | G | ρG |
| α² | G | ρG |
| ρ | ρG | Z |
| αρ | ρG | Z |
| α²ρ | ρG | Z |

TABLE I. Fusion rules for $\mathcal{H}_2$ and $\mathcal{H}_3$ (left) and module fusion rules (middle and right). The category $\mathcal{H}_3$ is rank 6, with a full subcategory, generated by $\{1, \alpha, \alpha^2\}$, equivalent to $\mathbf{Vec}(\mathbb{Z}/3\mathbb{Z})$. We define $X = \rho + \alpha\rho + \alpha^2\rho$, $Y = \Gamma + \alpha\Gamma + \alpha^2\Gamma + \Lambda$, and $Z = G + 3 \cdot \rho G$. Fusion rules for the modules were obtained from Ref. [9].

### Step 4.

Finally, to compute the $F$-symbols, we solve the linear equations

$$
\begin{matrix}
c & & & & & c & \\
b \diagdown\!\!\!\diagup e \!\!\!-\!\!\! d & = \sum_f \left[F_{abc}^d\right]_{ef} & b \diagup\!\!\!\!^f\!\!\!\diagdown \!\!\!-\!\!\! d \\
a & & & & & a &
\end{matrix} , \tag{51}
$$

where the vertices Eq. (50) are used. This gives

$$
\begin{array}{llll}
\left[F_{111}^1\right]_{11} = 1 & \left[F_{11\psi}^\psi\right]_{1\psi} = \frac{\omega_{11}^1}{\omega_{1\psi}^\psi} & \left[F_{1\psi1}^\psi\right]_{\psi\psi} = 1 & \left[F_{1\psi\psi}^1\right]_{\psi1} = \frac{\omega_{1\psi}^\psi}{\omega_{11}^1} \\
\left[F_{\psi11}^\psi\right]_{\psi1} = \frac{\omega_{\psi1}^1}{\omega_{11}^1} & \left[F_{\psi1\psi}^1\right]_{\psi\psi} = \frac{\omega_{\psi1}^1}{\omega_{1\psi}^\psi} & \left[F_{\psi\psi1}^1\right]_{1\psi} = \frac{\omega_{11}^1}{\omega_{\psi1}^1} & \left[F_{\psi\psi\psi}^\psi\right]_{11} = \frac{\omega_{1\psi}^\psi}{\omega_{\psi1}^1}
\end{array} , \tag{52}
$$

which is any true for any choice of $\omega_{ab}^x$, and corresponds to the fusion category $\mathbf{Rep}(\mathbb{Z}/2\mathbb{Z})$.

In the appendices, we provide similar worked examples for $(\mathbf{Vec}(S_3))_{\mathbf{Vec}}^*$ (Appendix B) and $(\mathbf{Rep}(S_3))_{\mathbf{Rep}(S_3)}^*$ (Appendix C), which explore some of the complications that arise in the more general case. Additionally, we supply $F$-symbols for $\mathcal{H}_1$, a category with multiplicity, in attached Mathematica files [14]. These were computed using the technique described here, using $(\mathcal{H}_3)_{\mathcal{M}_{3,1}}^*$, where the fusion category $\mathcal{H}_3$, and its module $\mathcal{M}_{3,1}$ are defined in Section V. These examples, including those in the attached code, illustrate the possible complications that can arise.

## V.  EXAMPLE: HAAGERUP FUSION CATEGORIES

A far more complicated application of our algorithm is finding the $F$-symbols of one of the Haagerup fusion categories. These categories originate from the Haagerup subfactor [27, 28], and they are of particular interest due to their outstanding role in the conjectured correspondence between subfactors and conformal field theories initially formulated by Vaughan Jones [29, 30]. Jones' conjecture states that for every unitary fusion category $\mathcal{C}$ (equivalently every finite depth subfactor), there is a conformal field theory (realized as a completely rational conformal net $\mathcal{A}$), such that $\mathcal{Z}(\mathcal{C}) \cong \mathbf{Rep}(\mathcal{A})$. We refer to Ref. [31] for a more complete exposition of the conjecture.

For subfactors with index less than four the conjecture is proven in Prop. 1.7 of Ref. [31], but the general case remains unproven. The first example above index four is the Haagerup subfactor, for which an associated CFT is yet to be proven, although it seems very likely that such a CFT exists [32].

The Morita equivalence class of fusion categories coming from the Haagerup subfactor contains three categories, commonly called $\mathcal{H}_1, \mathcal{H}_2, \mathcal{H}_3$, and their module categories were studied extensively in Ref. [9]. We take as the input category $\mathcal{H}_3$, with fusion rules given in Table I. The $F$-symbols for $\mathcal{H}_3$ were found in Refs. [11–13], and in an encoded form in Ref. [10]. We visualize the $F$-symbols in the left part of Fig. 1 in a gauge in which they are all real. For the category $\mathcal{H}_2$, $F$-symbols are also known [12, 13], leaving those for $\mathcal{H}_1$ the only unknown data.

Finding the $F$-symbols of $\mathcal{H}_1$ directly by solving the pentagon equation is considerably harder than the corresponding calculation for $\mathcal{H}_3$ due to the fact that $\mathcal{H}_1$ has multiplicities. It therefore makes sense to use the algorithm presented above to obtain the $F$-symbols for $\mathcal{H}_1$ via a module category over $\mathcal{H}_3$. We consider a rank 4 indecomposable module category over $\mathcal{H}_3$, which we refer to as $\mathcal{M}_{3,1}$. We obtained the fusion rules for this module from Ref. [9], although

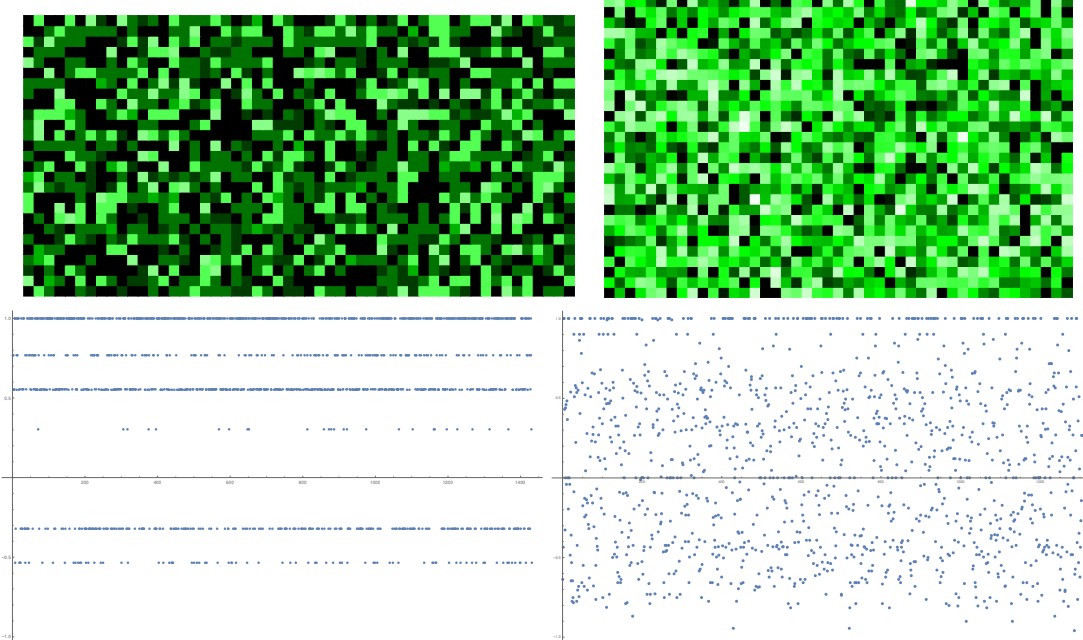

FIG. 1. A visualization of the $F$-symbol matrix elements in a unitary gauge, on the left is $\mathcal{H}_3$ [11–13], and on the right is $\mathcal{H}_1$. In the upper plot, white $= -1$, black $= +1$. All values are real.

they could be obtained more directly, for example by a brute-force search. We provide code for a tree-based search, inspired by a talk given by J. Slingerland [33], in the attached Mathematica file 'FindingModules.nb' [14]. The fusion rules are provided in the Table I.

Since these fusion rules are multiplicity free, it is reasonably easy to solve the module pentagon equation Eq. (14). The solution is provided in the attached file 'M31Data.m' [14], and can be verified and visualized in 'M31Data.nb' [14].

With this data obtained, one can apply the algorithm described above. The tube algebra (Eq. (21)) is 555 dimensional, so we refrain from extensive discussion of the computation. This algebra has 4 irreps, which we label $1, \mu, \eta, \nu$ following Ref. [9]. By forming the projectors onto the irreps, we can easily obtain fusion rules for these irreps

$$(\mathcal{H}_3)^*_{\mathcal{M}_{3,1}} = \begin{array}{c|cccc} \circ & 1 & \mu & \eta & \nu \\ \hline 1 & 1 & \mu & \eta & \nu \\ \mu & \mu & 1 + \nu & \eta + \nu & \eta + \mu + \nu \\ \eta & \eta & \eta + \nu & 1 + \eta + \mu + \nu & \eta + \mu + 2\nu \\ \nu & \nu & \eta + \mu + \nu & \eta + \mu + 2\nu & 1 + 2\eta + \mu + 2\nu \end{array} , \qquad (53)$$

which are the fusion rules of $\mathcal{H}_1$. Forming trivalent vertices, and using them to compute $F$-symbols gives the associator data for the Haagerup category $\mathcal{H}_1$. We provide a visualization in the right part of Fig. 1, and the numerical data in the attached file 'H1Data.m' [14]. In particular, these were obtained using the unitary version of the algorithm, and so are in a unitary gauge. Since there is multiplicity in the fusion rules, gauge freedom in these $F$-symbols is more than the phase freedom in $\mathcal{H}_3$. In the bottom right panel of Fig. 1, this is apparent since the values do not cluster, unlike the bottom left panel ($\mathcal{H}_3$). A full implementation of the algorithm used to obtain this data can be accessed from the notebook 'Main.nb' [14], which also allows this to be applied to the other examples in this manuscript.

To complete the Morita class, we include the data for $\mathcal{H}_2$ in the file 'DataH2.m' [14]. These data are obtained from a module category $\mathcal{M}_{3,2}$ (data in 'M32Data.m'). Unlike $\mathcal{M}_{3,1}$, this module has multiplicity three in its fusion rules as shown in the right panel of Table I. Since the module pentagon equations are quadratic, and due to the reduced gauge freedom, the module associator can be obtained more easily than would be possible for a fusion category with multiplicity three.

## VI. REMARKS

To summarize, we have described an algorithm to compute $F$-symbols for a fusion category $\mathcal{C}^*_{\mathcal{M}}$, given the data for a fusion category $\mathcal{C}$ and a module category $\mathcal{C} \curvearrowright \mathcal{M}$. By using a module version of the tube category, and its representations, this algorithm automates computation of these data.

To demonstrate the utility of our algorithm, we have applied it to obtain the $F$-symbols of the Haagerup Morita equivalence class, and in particular the category $\mathcal{H}_1$. This category has multiplicity in its fusion rules, which makes it extremely challenging to obtain the data by directly solving the pentagon equations. As such, this solution is among only a handful of such data that have been obtained for categories with multiplicity. Conversely, the module involved in the computation of $\mathcal{H}_2$ has multiplicity, however the simplified pentagon makes it possible to find the module associator directly. With this data in hand, the algorithm can be applied to find the data for the final category in the class.

The algorithm discussed in this manuscript requires as input data for one category in the Morita equivalence class. The problem of finding this initial data is the subject of a great deal of research, and the current algorithm provides no solution. In the case of a previously obtained solution, our algorithm allows for maximal use of the known data.

The only currently known 'exotic' fusion categories are those related to the 'extended Haagerup subfactor' [34] (EH). All other examples fall into some infinite family. For this reason, these are particularly interesting to study as this may aid in the classification. Previously, a complete understanding of the Morita equivalences has provided insight into the origins of purportedly exceptional fusion categories [35]. Although the categories Morita equivalent to EH are classified in Ref. [34], their data is not known. The algorithm presented here would greatly simplify the task of calculating the data. In the case that new fusion categories are discovered, it would also provide a way to more easily discover the Morita equivalences.

## ACKNOWLEDGMENTS

Research at Perimeter Institute is supported in part by the Government of Canada through the Department of Innovation, Science and Industry Canada and by the Province of Ontario through the Ministry of Colleges and Universities. R.W. acknowledges financial support from the National Centres of Competence in Research (NCCRs) QSIT (funded by the Swiss National Science Foundation under grant number 51NF40-185902) and *SwissMAP – The Mathematics of Physics*.

This work was initiated at the workshop "Fusion categories and tensor networks", hosted by the American Institute of Mathematics in March 2021. We thank AIM for their generosity.

We thank Corey Jones for enlightening discussions.

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

**Appendix A: Tensor structure on $\mathcal{C}_{\mathcal{M}}^*$**

Let $\mathcal{C}$ be a fusion category and $\mathcal{M}, \mathcal{N}$ be $\mathcal{C}$-modules. In this appendix, we briefly review the tensor structure on $\mathcal{C}_{\mathcal{M}}^*$. We will concentrate on recovering tube algebra data from this functor, but, given a tube algebra representation, the functor can be recovered similarly.

Let $F : \mathcal{M} \to \mathcal{N}$ be a module functor. This data specifies a vector space $\mathcal{N}(F(m), n)$, which we denote

$$\mathcal{N}(F(m), n) \leftrightarrow \qquad . \tag{A1}$$

After a basis has been chosen, vectors in $\mathcal{N}(F(m), n)$ are indicated by

$$ . \tag{A2}$$

The tube algebra action

$$x \left( \quad \bullet\, v_i \quad = \sum_j a_{ij}\, \bullet\, w_j \right. \tag{A3}$$

on this vector space is extracted from the functor $F$ as follows:

$$\mathcal{N}(F(m_1), n_1) \xrightarrow{x \triangleright -} \mathcal{N}(x \triangleright F(m_1), x \triangleright n_1) \xrightarrow{\text{postcompose with } \beta}$$
$$\mathcal{N}(x \triangleright F(m_1), n_2) \xrightarrow{F \text{ coherence iso.}} \mathcal{N}(F(x \triangleright m_1), n_2) \xrightarrow{\text{precompose with } F(\alpha)} \mathcal{N}(F(m_2), n_2) \tag{A4}$$

Now consider the composition of two tensor functors $F : \mathcal{M} \to \mathcal{N}$ and $G : \mathcal{N} \to \mathcal{P}$. The coherence isomorphism for

$$F \otimes G := G \circ F \tag{A5}$$

is given by

$$x \triangleright G(F(m)) \xrightarrow{G \text{ coherence iso.}} G(x \triangleright F(m)) \xrightarrow{F \text{ coherence iso.}} G(F(x \triangleright m)) \tag{A6}$$

Now assume that $x \triangleright F(m) \cong n$ and choose trivalent vertices $\gamma : x \triangleright F(m) \to n$ and $\delta : n \to a \triangleright F(m)$ realizing this isomorphism. We can decompose the coherence isomorphism for $G \circ F$ as

$$a \triangleright G(F(m)) \xrightarrow{G \text{ coherence iso.}} G(x \triangleright F(m)) \xrightarrow{G(\gamma)} G(n) \xrightarrow{G(\delta)} G(x \triangleright F(m)) \xrightarrow{F \text{ coherence iso.}} G(F(x \triangleright m)) \tag{A7}$$

Substituting Eq. (A7) into Eq. (A4) tells us that the tube action for $G \circ F$ can be interpreted as

$$ \rightarrow \tag{A8}$$

followed by applying the $F$ and $G$ actions independently. Notice that this is like a higher categorical version of the well known group theory trick $xy = xgg^{-1}y$.

## Appendix B: Worked Example: $\mathbf{Vec}(S_3)$

We work through another relatively simple example to recover the $F$-symbols of $\mathbf{Rep}(S_3)$ from a module over $\mathbf{Vec}(S_3)$. This example has a two-dimensional representation, so is slightly more complicated than that in Section IV. Additionally, since $\mathbf{Rep}(S_3)$ is not equivalent as a fusion category to $\mathbf{Vec}(S_3)$, it is perhaps more clear that the output $F$-symbols are distinct from the input. Conversely, since there is a single simple object in the module category, computations within this example remain straightforward. In particular, all boundary tubes can be stacked to give a nonzero picture.

The group $S_3$ is given by the presentation

$$S_3 = \langle \sigma, \tau \,|\, \sigma^3 = \tau^2 = (\sigma\tau)^2 = 1 \rangle. \tag{B1}$$

The simple objects of $\mathbf{Vec}(S_3)$ are labeled by the group elements, with fusion given by group multiplication. All $F$-symbols are 1 when permitted by fusion. In all cases, we neglect to draw the strings corresponding to the unit object 1.

We consider a module category $\mathcal{M}$ with a single simple object, denoted $*$ or a blue string. All module $L-$symbols are 1 when permitted by fusion.

The boundary tube algebra is 6-dimensional, with picture basis

$$\Lambda = \left\{ \mathbf{T}_1 = \quad , \mathbf{T}_\sigma = \sigma \quad , \mathbf{T}_{\sigma^2} = \sigma^2 \quad , \mathbf{T}_\tau = \tau \quad , \mathbf{T}_{\sigma\tau} = \sigma\tau \quad , \mathbf{T}_{\sigma^2\tau} = \sigma^2\tau \quad \right\}. \tag{B2}$$

The multiplication on the tubes is given by group multiplication on their label $\mathbf{T}_i\,\mathbf{T}_j = \mathbf{T}_{i\cdot j}$.

### Step 1.

The tube algebra composes into two 1-dimensional algebras and one 2-dimensional algebra: $\mathbb{C}[S_3] \cong \mathbb{C} \oplus \mathbb{C} \oplus M_2(\mathbb{C})$. A complete set of matrix units is given by

$$[e_1]_{00} = \frac{\mathbf{T}_1 + \mathbf{T}_\sigma + \mathbf{T}_{\sigma^2} + \mathbf{T}_\tau + \mathbf{T}_{\sigma\tau} + \mathbf{T}_{\sigma^2\tau}}{6} = \frac{1}{6}\left( \quad + \sigma \quad + \sigma^2 \quad + \tau \quad + \sigma\tau \quad + \sigma^2\tau \quad \right), \tag{B3a}$$

$$[e_\psi]_{00} = \frac{\mathbf{T}_1 + \mathbf{T}_\sigma + \mathbf{T}_{\sigma^2} - \mathbf{T}_\tau - \mathbf{T}_{\sigma\tau} - \mathbf{T}_{\sigma^2\tau}}{6} = \frac{1}{6}\left( \quad + \sigma \quad + \sigma^2 \quad - \tau \quad - \sigma\tau \quad - \sigma^2\tau \quad \right), \tag{B3b}$$

$$[e_\pi]_{00} = \frac{2\,\mathbf{T}_1 - \mathbf{T}_\sigma - \mathbf{T}_{\sigma^2} - 2\,\mathbf{T}_\tau + \mathbf{T}_{\sigma\tau} + \mathbf{T}_{\sigma^2\tau}}{6} = \frac{1}{6}\left( 2 \quad - \sigma \quad - \sigma^2 \quad - 2 \tau \quad + \sigma\tau \quad + \sigma^2\tau \quad \right), \tag{B3c}$$

$$[e_\pi]_{01} = \frac{-\mathbf{T}_\sigma + \mathbf{T}_{\sigma^2} - \mathbf{T}_{\sigma\tau} + \mathbf{T}_{\sigma^2\tau}}{2\sqrt{3}} = \frac{1}{2\sqrt{3}}\left( - \sigma \quad + \sigma^2 \quad - \sigma\tau \quad + \sigma^2\tau \quad \right), \tag{B3d}$$

$$[e_\pi]_{10} = \frac{\mathbf{T}_\sigma - \mathbf{T}_{\sigma^2} - \mathbf{T}_{\sigma\tau} + \mathbf{T}_{\sigma^2\tau}}{2\sqrt{3}} = \frac{1}{2\sqrt{3}}\left( \sigma \quad - \sigma^2 \quad - \sigma\tau \quad + \sigma^2\tau \quad \right), \tag{B3e}$$

$$[e_\pi]_{11} = \frac{2\,\mathbf{T}_1 - \mathbf{T}_\sigma - \mathbf{T}_{\sigma^2} + 2\,\mathbf{T}_\tau - \mathbf{T}_{\sigma\tau} - \mathbf{T}_{\sigma^2\tau}}{6} = \frac{1}{6}\left( 2 \quad - \sigma \quad - \sigma^2 \quad + 2 \tau \quad - \sigma\tau \quad - \sigma^2\tau \quad \right). \tag{B3f}$$

A basis for the representations is given by

$$[v_1]_0 = [e_1]_{00}, \qquad [v_\psi]_0 = [e_\psi]_{00}, \qquad [v_\pi]_0 = [e_\pi]_{00}, \qquad [v_\pi]_1 = [e_\pi]_{10}, \qquad \text{(B4)}$$

with

$$[e_x]_{i,j}[v_y]_k = \delta_x^y \delta_j^k [v_x]_i. \qquad \text{(B5)}$$

Using the central idempotents

$$\mathbb{1}_1 := [e_1]_{00} \qquad \text{(B6a)}$$
$$\mathbb{1}_\psi := [e_\psi]_{00} \qquad \text{(B6b)}$$
$$\mathbb{1}_\pi := [e_\pi]_{00} + [e_\pi]_{11}, \qquad \text{(B6c)}$$

we can project onto a given representation. This is useful for computing the fusion rules for $\mathcal{C}_\mathcal{M}^*$, but the full representations are required to compute the $F$-symbols.

The fusion category $\mathcal{C}_\mathcal{M}^*$ therefore has 3 simple objects, labeled $1, \psi, \pi$.

**Step 2.**

The composite basis is 36-dimensional, with picture basis

$$\left\{ \begin{array}{c} \beta \\ \alpha \end{array} \middle| \; \alpha, \beta \in S_3 \right\}. \qquad \text{(B7)}$$

The tensor products of the representations $[v_x]$ form a 16-dimensional subspace with basis $[v_x]_i \otimes [v_y]_j$.

**Step 3.**

To find the required trivalent vertices, we need to decompose the composite space. This can be achieved as follows:

- Fix $x, y$ representations. Pick a generic vector $V = \sum_{i,j} \alpha_{x,i}^{y,j} [v_x]_i \otimes [v_y]_j$,

- For each representation $z$, apply $[e_z]_{00}$, giving a new vector $V^{(z)}$.

- If $V^{(z)} = 0$, the representation $z$ does not occur inside the tensor product $x \otimes y$.

- The vector $V^{(z)} \neq 0$ will have at least one free parameter $\alpha$. If it has exactly one, it can be fixed to any value. Ultimately, this corresponds to a choice of gauge for the trivalent vertices. If there are multiple free parameters ($n$ of them), $z$ occurs multiple times within $x \otimes y$. In that case, form $n$ linearly independent vectors $V^{(z,q)}$ with all but one of the free parameters set to 0, and the remaining one fixed, for example, to 1.

- We can now identify $[v_z]_0$ with $V^{(z,q)}$ for each $q \in \{0, \ldots, n-1\}$ since $[e_z]_{00} V^{(z,q)} = V^{(z,q)}$. Fill out the representations by applying $[e_z]_{i0}$, giving vectors $V_i^{(z,q)}$.

- The matrix elements of the trivalent vertex

$$\begin{array}{c} y \\ \diagdown \\ x \diagup \\ q \end{array} \!\!\! z \qquad \text{(B8)}$$

are given by the coefficients of $[v_x]_i \otimes [v_y]_j$ in the vector $V_k^{(z,q)}$, where the rows are labeled by $\left\{ [v_x]_i \otimes [v_y]_j \right\}_{i,j}$, and the columns by $\left\{ V_k^{(z,q)} \right\}_k$.

For the present example, there are no multiplicities. For clarity, we leave the free parameters unfixed, naming them $\omega$. This serves to demonstrate that they correspond to the gauge freedom in the $F$-symbols. The trivalent vertices are given by the matrices

$$
\begin{array}{c} 1 \\ 1 \end{array}\!\!\!\succ\!\!\!-1 \;=\; [v_1]_0 \otimes [v_1]_0 \begin{pmatrix} [v_1]_0 \\ \omega_{11}^1 \end{pmatrix} \times 6^{1/4}
\qquad
\begin{array}{c} \psi \\ 1 \end{array}\!\!\!\succ\!\!\!-\psi \;=\; [v_1]_0 \otimes [v_\psi]_0 \begin{pmatrix} [v_\psi]_0 \\ \omega_{1\psi}^\psi \end{pmatrix} \times 6^{1/4}
$$

$$
\begin{array}{c} 1 \\ \psi \end{array}\!\!\!\succ\!\!\!-\psi \;=\; [v_\psi]_0 \otimes [v_1]_0 \begin{pmatrix} [v_\psi]_0 \\ \omega_{\psi 1}^\psi \end{pmatrix} \times 6^{1/4}
\qquad
\begin{array}{c} \psi \\ \psi \end{array}\!\!\!\succ\!\!\!-1 \;=\; [v_\psi]_0 \otimes [v_\psi]_0 \begin{pmatrix} [v_1]_0 \\ \omega_{\psi\psi}^1 \end{pmatrix} \times 6^{1/4}
$$

$$
\begin{array}{c} \pi \\ 1 \end{array}\!\!\!\succ\!\!\!-\pi \;=\; \begin{array}{c} [v_1]_0 \otimes [v_\pi]_0 \\ [v_1]_0 \otimes [v_\pi]_1 \end{array}\begin{pmatrix} [v_\pi]_0 & [v_\pi]_1 \\ \omega_{1\pi}^\pi & 0 \\ 0 & \omega_{1\pi}^\pi \end{pmatrix} \times 6^{1/4}
\qquad
\begin{array}{c} 1 \\ \pi \end{array}\!\!\!\succ\!\!\!-\pi \;=\; \begin{array}{c} [v_\pi]_0 \otimes [v_1]_0 \\ [v_\pi]_1 \otimes [v_1]_0 \end{array}\begin{pmatrix} [v_\pi]_0 & [v_\pi]_1 \\ \omega_{\pi 1}^\pi & 0 \\ 0 & \omega_{\pi 1}^\pi \end{pmatrix} \times 6^{1/4}
$$

$$
\begin{array}{c} \pi \\ \psi \end{array}\!\!\!\succ\!\!\!-\pi \;=\; \begin{array}{c} [v_\psi]_0 \otimes [v_\pi]_0 \\ [v_\psi]_0 \otimes [v_\pi]_1 \end{array}\begin{pmatrix} [v_\pi]_0 & [v_\pi]_1 \\ 0 & -\omega_{\psi\pi}^\pi \\ \omega_{\psi\pi}^\pi & 0 \end{pmatrix} \times 6^{1/4}
\qquad
\begin{array}{c} \psi \\ \pi \end{array}\!\!\!\succ\!\!\!-\pi \;=\; \begin{array}{c} [v_\pi]_0 \otimes [v_\psi]_0 \\ [v_\pi]_1 \otimes [v_\psi]_0 \end{array}\begin{pmatrix} [v_\pi]_0 & [v_\pi]_1 \\ 0 & -\omega_{\pi\psi}^\pi \\ \omega_{\pi\psi}^\pi & 0 \end{pmatrix} \times 6^{1/4}
$$

$$
\begin{array}{c} \pi \\ \pi \end{array}\!\!\!\succ\!\!\!-1 \;=\; \begin{array}{c} [v_\pi]_0 \otimes [v_\pi]_0 \\ [v_\pi]_0 \otimes [v_\pi]_1 \\ [v_\pi]_1 \otimes [v_\pi]_0 \\ [v_\pi]_1 \otimes [v_\pi]_1 \end{array}\begin{pmatrix} [v_1]_0 \\ \omega_{\pi\pi}^1 \\ 0 \\ 0 \\ \omega_{\pi\pi}^1 \end{pmatrix} \times \left(\frac{3}{32}\right)^{1/4}
\qquad
\begin{array}{c} \pi \\ \pi \end{array}\!\!\!\succ\!\!\!-\psi \;=\; \begin{array}{c} [v_\pi]_0 \otimes [v_\pi]_0 \\ [v_\pi]_0 \otimes [v_\pi]_1 \\ [v_\pi]_1 \otimes [v_\pi]_0 \\ [v_\pi]_1 \otimes [v_\pi]_1 \end{array}\begin{pmatrix} [v_\psi]_0 \\ 0 \\ \omega_{\pi\pi}^\psi \\ -\omega_{\pi\pi}^\psi \\ 0 \end{pmatrix} \times \left(\frac{3}{32}\right)^{1/4}
$$

$$
\begin{array}{c} \pi \\ \pi \end{array}\!\!\!\succ\!\!\!-\pi \;=\; \begin{array}{c} [v_\pi]_0 \otimes [v_\pi]_0 \\ [v_\pi]_0 \otimes [v_\pi]_1 \\ [v_\pi]_1 \otimes [v_\pi]_0 \\ [v_\pi]_1 \otimes [v_\pi]_1 \end{array}\begin{pmatrix} [v_\pi]_0 & [v_\pi]_1 \\ 0 & \omega_{\pi\pi}^\pi \\ \omega_{\pi\pi}^\pi & 0 \\ \omega_{\pi\pi}^\pi & 0 \\ 0 & -\omega_{\pi\pi}^\pi \end{pmatrix} \times \left(\frac{3}{8}\right)^{1/4}.
$$

**Step 4.**

The remainder of the calculation is straightforward linear algebra, solving the linear equations

$$
\begin{array}{c} c \\ b \\ a \end{array}\!\!\!\succ\!\!\!\underset{e}{\succ}\!\!\!-d \;=\; \sum_f \left[F_{abc}^d\right]_{ef} \begin{array}{c} c \\ b \\ a \end{array}\!\!\!\succ\!\!\!\overset{f}{\succ}\!\!\!-d \;, \tag{B9}
$$

where joined indices corresponds to (conventional) tensor contraction of the reshaped matrices. For example

$$
\begin{array}{c} \pi \\ \pi \\ \pi \end{array}\!\!\!\succ\!\!\!\underset{\pi}{\succ}\!\!\!-\pi \;=\; \begin{pmatrix} 1 & 0 \\ 0 & -1 \\ 0 & 1 \\ 1 & 0 \\ 0 & 1 \\ 1 & 0 \\ -1 & 0 \\ 0 & 1 \end{pmatrix} \times \sqrt{\frac{3}{8}}(\omega_{\pi\pi}^\pi)^2, \tag{B10}
$$

and

$$
\pi \,\pi \,\pi \searrow^{1} \!\!\!\!- \pi \;=\; \begin{pmatrix} 1 & 0 \\ 0 & 0 \\ 0 & 0 \\ 1 & 0 \\ 0 & 1 \\ 0 & 0 \\ 0 & 0 \\ 0 & 1 \end{pmatrix} \times \sqrt{\tfrac{3}{4}}\,\omega^{\pi}_{\pi 1}\omega^{1}_{\pi\pi}
\tag{B11a}
$$

$$
\pi \,\pi \,\pi \searrow^{\psi} \!\!\!\!- \pi \;=\; \begin{pmatrix} 0 & 0 \\ 0 & -1 \\ 0 & 1 \\ 0 & 0 \\ 0 & 0 \\ 1 & 0 \\ -1 & 0 \\ 0 & 0 \end{pmatrix} \times \sqrt{\tfrac{3}{4}}\,\omega^{\pi}_{\pi\psi}\omega^{\psi}_{\pi\pi}
\tag{B11b}
$$

$$
\pi \,\pi \,\pi \searrow^{\pi} \!\!\!\!- \pi \;=\; \begin{pmatrix} 1 & 0 \\ 0 & 1 \\ 0 & 1 \\ -1 & 0 \\ 0 & -1 \\ 1 & 0 \\ 1 & 0 \\ 0 & 1 \end{pmatrix} \times \sqrt{\tfrac{3}{8}}\,(\omega^{\pi}_{\pi\pi})^{2}.
\tag{B11c}
$$

From this, we read off that

$$
\left[F^{\pi}_{\pi\pi\pi}\right]_{\pi 1} = \frac{\left(\omega^{\pi}_{\pi,\pi}\right)^{2}}{\sqrt{2}\,\omega^{\pi}_{\pi,1}\omega^{1}_{\pi,\pi}}
\qquad
\left[F^{\pi}_{\pi\pi\pi}\right]_{\pi\psi} = \frac{\left(\omega^{\pi}_{\pi,\pi}\right)^{2}}{\sqrt{2}\,\omega^{\pi}_{\pi,\psi}\omega^{\psi}_{\pi,\pi}}
\qquad
\left[F^{\pi}_{\pi\pi\pi}\right]_{\pi\pi} = 0.
\tag{B12}
$$

The full set of $F$-symbols is computed similarly. Those that are not required to be zero by fusion are

$$
\begin{array}{lllll}
\left[F^{1}_{111}\right]_{11} = 1 &
\left[F^{\psi}_{11\psi}\right]_{1\psi} = \frac{\omega^{1}_{11}}{\omega^{\psi}_{1\psi}} &
\left[F^{\pi}_{11\pi}\right]_{1\pi} = \frac{\omega^{1}_{11}}{\omega^{\pi}_{1\pi}} &
\left[F^{\psi}_{1\psi 1}\right]_{\psi\psi} = 1 &
\left[F^{1}_{1\psi\psi}\right]_{\psi 1} = \frac{\omega^{\psi}_{1\psi}}{\omega^{1}_{11}} \\[6pt]
\left[F^{\pi}_{1\psi\pi}\right]_{\psi\pi} = \frac{\omega^{\psi}_{1\psi}}{\omega^{\pi}_{1\pi}} &
\left[F^{\pi}_{1\pi 1}\right]_{\pi\pi} = 1 &
\left[F^{\pi}_{1\pi\psi}\right]_{\pi\pi} = 1 &
\left[F^{1}_{1\pi\pi}\right]_{\pi 1} = \frac{\omega^{\pi}_{1\pi}}{\omega^{1}_{11}} &
\left[F^{\psi}_{1\pi\pi}\right]_{\pi\psi} = \frac{\omega^{\pi}_{1\pi}}{\omega^{\psi}_{1\psi}} \\[6pt]
\left[F^{\pi}_{1\pi\pi}\right]_{\pi\pi} = 1 &
\left[F^{\psi}_{\psi 11}\right]_{\psi 1} = \frac{\omega^{\psi}_{\psi 1}}{\omega^{1}_{11}} &
\left[F^{1}_{\psi 1\psi}\right]_{\psi\psi} = \frac{\omega^{\psi}_{\psi 1}}{\omega^{\psi}_{1\psi}} &
\left[F^{\pi}_{\psi 1\pi}\right]_{\psi\pi} = \frac{\omega^{\psi}_{\psi 1}}{\omega^{\pi}_{1\pi}} &
\left[F^{1}_{\psi\psi 1}\right]_{1\psi} = \frac{\omega^{1}_{11}}{\omega^{\psi}_{\psi 1}} \\[6pt]
\left[F^{\psi}_{\psi\psi\psi}\right]_{11} = \frac{\omega^{\psi}_{1\psi}}{\omega^{\psi}_{\psi 1}} &
\left[F^{\pi}_{\psi\psi\pi}\right]_{1\pi} = -\frac{\omega^{\pi}_{1\pi}\omega^{1}_{\psi\psi}}{\omega^{\pi}_{\psi\pi}{}^{2}} &
\left[F^{\pi}_{\psi\pi 1}\right]_{\pi\pi} = 1 &
\left[F^{\pi}_{\psi\pi\psi}\right]_{\pi\pi} = 1 &
\left[F^{\psi}_{\psi\pi\pi}\right]_{\pi 1} = \frac{\omega^{\psi}_{\pi\pi}\omega^{\pi}_{\psi\pi}}{\omega^{1}_{\pi\pi}\omega^{\psi}_{\psi 1}} \\[6pt]
\left[F^{1}_{\psi\pi\pi}\right]_{\pi\psi} = -\frac{\omega^{\pi}_{\pi\pi}\omega^{\pi}_{\psi\pi}}{\omega^{\psi}_{\pi\pi}\omega^{1}_{\psi\psi}} &
\left[F^{\pi}_{\psi\pi\pi}\right]_{\pi\pi} = -1 &
\left[F^{\pi}_{\pi 11}\right]_{\pi 1} = \frac{\omega^{\pi}_{\pi 1}}{\omega^{1}_{11}} &
\left[F^{\pi}_{\pi 1\psi}\right]_{\pi\psi} = \frac{\omega^{\pi}_{\pi 1}}{\omega^{\psi}_{1\psi}} &
\left[F^{1}_{\pi 1\pi}\right]_{\pi\pi} = \frac{\omega^{\pi}_{\pi 1}}{\omega^{\pi}_{1\pi}} \\[6pt]
\left[F^{\psi}_{\pi 1\pi}\right]_{\pi\pi} = \frac{\omega^{\pi}_{\pi 1}}{\omega^{\pi}_{1\pi}} &
\left[F^{\pi}_{\pi 1\pi}\right]_{\pi\pi} = \frac{\omega^{\pi}_{\pi 1}}{\omega^{\pi}_{1\pi}} &
\left[F^{\pi}_{\pi\psi 1}\right]_{\pi\psi} = \frac{\omega^{\pi}_{\pi 1}}{\omega^{\psi}_{\psi 1}} &
\left[F^{\pi}_{\pi\psi\psi}\right]_{\pi 1} = -\frac{\omega^{\pi}_{\pi\psi}{}^{2}}{\omega^{\pi}_{\pi 1}\omega^{1}_{\psi\psi}} &
\left[F^{1}_{\pi\psi\pi}\right]_{\pi\pi} = -\frac{\omega^{\pi}_{\pi\psi}}{\omega^{\pi}_{\psi\pi}} \\[6pt]
\left[F^{\psi}_{\pi\psi\pi}\right]_{\pi\pi} = -\frac{\omega^{\pi}_{\pi\psi}}{\omega^{\pi}_{\psi\pi}} &
\left[F^{\pi}_{\pi\psi\pi}\right]_{\pi\pi} = \frac{\omega^{\pi}_{\pi\psi}}{\omega^{\pi}_{\psi\pi}} &
\left[F^{1}_{\pi\pi 1}\right]_{1\pi} = \frac{\omega^{1}_{11}}{\omega^{\pi}_{\pi 1}} &
\left[F^{\psi}_{\pi\pi 1}\right]_{\psi\pi} = \frac{\omega^{\psi}_{\psi 1}}{\omega^{\pi}_{\pi 1}} &
\left[F^{\pi}_{\pi\pi 1}\right]_{\pi\pi} = 1 \\[6pt]
\left[F^{\psi}_{\pi\pi\psi}\right]_{1\pi} = -\frac{\omega^{\psi}_{1\psi}\omega^{1}_{\pi\pi}}{\omega^{\psi}_{\pi\pi}\omega^{\pi}_{\pi\psi}} &
\left[F^{1}_{\pi\pi\psi}\right]_{\psi\pi} = \frac{\omega^{\psi}_{\pi\pi}\omega^{1}_{\psi\psi}}{\omega^{1}_{\pi\pi}\omega^{\pi}_{\pi\psi}} &
\left[F^{\pi}_{\pi\pi\psi}\right]_{\pi\pi} = -1 &
\left[F^{\pi}_{\pi\pi\pi}\right]_{11} = \frac{\omega^{\pi}_{1\pi}}{2\omega^{\pi}_{\pi 1}} &
\left[F^{\pi}_{\pi\pi\pi}\right]_{1\psi} = -\frac{\omega^{\pi}_{1\pi}\omega^{\pi}_{\pi\pi}}{2\omega^{\psi}_{\pi\pi}\omega^{\pi}_{\pi\psi}} \\[6pt]
\left[F^{\pi}_{\pi\pi\pi}\right]_{1\pi} = \frac{\omega^{\pi}_{1\pi}\omega^{1}_{\pi\pi}}{\sqrt{2}\,\omega^{\pi}_{\pi 1}{}^{2}} &
\left[F^{\pi}_{\pi\pi\pi}\right]_{\psi 1} = \frac{\omega^{\psi}_{\pi\pi}\omega^{\pi}_{\psi\pi}}{2\omega^{\pi}_{\pi 1}\omega^{1}_{\pi\pi}} &
\left[F^{\pi}_{\pi\pi\pi}\right]_{\psi\psi} = -\frac{\omega^{\pi}_{\pi\pi}}{2\omega^{\pi}_{\pi\psi}} &
\left[F^{\pi}_{\pi\pi\pi}\right]_{\psi\pi} = -\frac{\omega^{\psi}_{\pi\pi}\omega^{\pi}_{\psi\pi}}{\sqrt{2}\,\omega^{\pi}_{\pi\pi}{}^{2}} &
\left[F^{\pi}_{\pi\pi\pi}\right]_{\pi 1} = \frac{\omega^{\pi}_{\pi\pi}{}^{2}}{\sqrt{2}\,\omega^{\pi}_{\pi 1}\omega^{1}_{\pi\pi}} \\[6pt]
\left[F^{\pi}_{\pi\pi\pi}\right]_{\pi\psi} = \frac{\omega^{\pi}_{\pi\pi}{}^{2}}{\sqrt{2}\,\omega^{\psi}_{\pi\pi}\omega^{\pi}_{\pi\psi}} &
\left[F^{1}_{\pi\pi\pi}\right]_{\pi\pi} = 1 &
\left[F^{\psi}_{\pi\pi\pi}\right]_{\pi\pi} = -1 &
\left[F^{\pi}_{\pi\pi\pi}\right]_{\pi\pi} = 0 &
\end{array}
$$

It can readily be verified that these obey the pentagon equations, and are the $F$-symbols, up to a choice of gauge, for $\mathbf{Rep}(S_3)$ as expected.

$$\left[F^{1}_{111}\right]_{11} = 1 \quad \left[F^{\psi}_{11\psi}\right]_{1\psi} = 1 \quad \left[F^{\pi}_{11\pi}\right]_{1\pi} = 1 \quad \left[F^{\psi}_{1\psi 1}\right]_{\psi\psi} = 1 \quad \left[F^{1}_{1\psi\psi}\right]_{\psi 1} = 1 \quad \left[F^{\pi}_{1\psi\pi}\right]_{\psi\pi} = 1 \quad \left[F^{\pi}_{1\pi 1}\right]_{\pi\pi} = 1$$

$$\left[F^{\pi}_{1\pi\psi}\right]_{\pi\pi} = 1 \quad \left[F^{1}_{1\pi\pi}\right]_{\pi 1} = 1 \quad \left[F^{\psi}_{1\pi\pi}\right]_{\pi\psi} = 1 \quad \left[F^{\pi}_{1\pi\pi}\right]_{\pi\pi} = 1 \quad \left[F^{\psi}_{\psi 11}\right]_{\psi 1} = 1 \quad \left[F^{1}_{\psi 1\psi}\right]_{\psi\psi} = 1 \quad \left[F^{\pi}_{\psi 1\pi}\right]_{\psi\pi} = 1$$

$$\left[F^{1}_{\psi\psi 1}\right]_{1\psi} = 1 \quad \left[F^{\psi}_{\psi\psi\psi}\right]_{11} = 1 \quad \left[F^{\pi}_{\psi\psi\pi}\right]_{1\pi} = 1 \quad \left[F^{\pi}_{\psi\pi 1}\right]_{\pi\pi} = 1 \quad \left[F^{\pi}_{\psi\pi\psi}\right]_{\pi\pi} = 1 \quad \left[F^{\psi}_{\psi\pi\pi}\right]_{\pi 1} = 1 \quad \left[F^{1}_{\psi\pi\pi}\right]_{\pi\psi} = 1$$

$$\left[F^{\pi}_{\psi\pi\pi}\right]_{\pi\pi} = -1 \quad \left[F^{\pi}_{\pi 11}\right]_{\pi 1} = 1 \quad \left[F^{\pi}_{\pi 1\psi}\right]_{\pi\psi} = 1 \quad \left[F^{1}_{\pi 1\pi}\right]_{\pi\pi} = 1 \quad \left[F^{\psi}_{\pi 1\pi}\right]_{\pi\pi} = 1 \quad \left[F^{\pi}_{\pi 1\pi}\right]_{\pi\pi} = 1 \quad \left[F^{\pi}_{\pi\psi 1}\right]_{\pi\psi} = 1$$

$$\left[F^{\pi}_{\pi\psi\psi}\right]_{\pi 1} = 1 \quad \left[F^{1}_{\pi\psi\pi}\right]_{\pi\pi} = 1 \quad \left[F^{\psi}_{\pi\psi\pi}\right]_{\pi\pi} = 1 \quad \left[F^{\pi}_{\pi\psi\pi}\right]_{\pi\pi} = -1 \quad \left[F^{1}_{\pi\pi 1}\right]_{1\pi} = 1 \quad \left[F^{\psi}_{\pi\pi 1}\right]_{\psi\pi} = 1 \quad \left[F^{\pi}_{\pi\pi 1}\right]_{\pi\pi} = 1$$

$$\left[F^{\psi}_{\pi\pi\psi}\right]_{1\pi} = 1 \quad \left[F^{1}_{\pi\pi\psi}\right]_{\psi\pi} = 1 \quad \left[F^{\pi}_{\pi\pi\psi}\right]_{\pi\pi} = -1 \quad \left[F^{\pi}_{\pi\pi\pi}\right]_{11} = \tfrac{1}{2} \quad \left[F^{\pi}_{\pi\pi\pi}\right]_{1\psi} = \tfrac{1}{2} \quad \left[F^{\pi}_{\pi\pi\pi}\right]_{1\pi} = \tfrac{1}{\sqrt{2}} \quad \left[F^{\pi}_{\pi\pi\pi}\right]_{\psi 1} = \tfrac{1}{2}$$

$$\left[F^{\pi}_{\pi\pi\pi}\right]_{\psi\psi} = \tfrac{1}{2} \quad \left[F^{\pi}_{\pi\pi\pi}\right]_{\psi\pi} = -\tfrac{1}{\sqrt{2}} \quad \left[F^{\pi}_{\pi\pi\pi}\right]_{\pi 1} = \tfrac{1}{\sqrt{2}} \quad \left[F^{\pi}_{\pi\pi\pi}\right]_{\pi\psi} = -\tfrac{1}{\sqrt{2}} \quad \left[F^{1}_{\pi\pi\pi}\right]_{\pi\pi} = 1 \quad \left[F^{\psi}_{\pi\pi\pi}\right]_{\pi\pi} = -1 \quad \left[F^{\pi}_{\pi\pi\pi}\right]_{\pi\pi} = 0$$

## Appendix C: Worked Example: $\mathbf{Rep}(S_3)$

We work through another relatively simple example to recover the $F$-symbols of $\mathbf{Rep}(S_3)$ from a module over $\mathbf{Rep}(S_3)$. This example is slightly more complicated than that in Appendix B since the module category we choose has 3 simple objects. Because of this, and unlike the previous two examples, not all boundary diagrams are valid.

As input, we use the $F$-symbols computed in Appendix B. We consider $\mathbf{Rep}(S_3)$ as a module over itself, so the module category $\mathcal{M}$ also has 3 simple objects.

The boundary tube algebra is $3^2 + 3^2 + 5^2 = 43$ dimensional, where the decomposition foreshadows the decomposition into irreducible representations. Defining

$$\mathbf{T}[ab|cd]_x := \raisebox{-2em}{\includegraphics{placeholder}} \quad x \begin{array}{c} d \\ c \\ b \\ a \end{array}, \tag{C1}$$

the picture basis is

$$\Lambda = \left\{ \begin{array}{llllllll} \mathbf{T}[11|11]_1, & \mathbf{T}[11|\psi\psi]_\psi, & \mathbf{T}[11|\pi\pi]_\pi, & \mathbf{T}[1\psi|1\psi]_1, & \mathbf{T}[1\psi|\psi1]_\psi, & \mathbf{T}[1\psi|\pi\pi]_\pi, & \mathbf{T}[1\pi|1\pi]_1, & \mathbf{T}[1\pi|\psi\pi]_\psi, \\ \mathbf{T}[1\pi|\pi1]_\pi, & \mathbf{T}[1\pi|\pi\psi]_\pi, & \mathbf{T}[1\pi|\pi\pi]_\pi, & \mathbf{T}[\psi1|1\psi]_\psi, & \mathbf{T}[\psi1|\psi1]_1, & \mathbf{T}[\psi1|\pi\pi]_\pi, & \mathbf{T}[\psi\psi|11]_\psi, & \mathbf{T}[\psi\psi|\psi\psi]_1, \\ \mathbf{T}[\psi\psi|\pi\pi]_\pi, & \mathbf{T}[\psi\pi|1\pi]_\psi, & \mathbf{T}[\psi\pi|\psi\pi]_1, & \mathbf{T}[\psi\pi|\pi1]_\pi, & \mathbf{T}[\psi\pi|\pi\psi]_\pi, & \mathbf{T}[\psi\pi|\pi\pi]_\pi, & \mathbf{T}[\pi1|1\pi]_\pi, & \mathbf{T}[\pi1|\psi\pi]_\pi, \\ \mathbf{T}[\pi1|\pi1]_1, & \mathbf{T}[\pi1|\pi\psi]_\psi, & \mathbf{T}[\pi1|\pi\pi]_\pi, & \mathbf{T}[\pi\psi|1\pi]_\pi, & \mathbf{T}[\pi\psi|\psi\pi]_\pi, & \mathbf{T}[\pi\psi|\pi1]_\psi, & \mathbf{T}[\pi\psi|\pi\psi]_1, & \mathbf{T}[\pi\psi|\pi\pi]_\pi, \\ \mathbf{T}[\pi\pi|11]_\pi, & \mathbf{T}[\pi\pi|1\psi]_\pi, & \mathbf{T}[\pi\pi|1\pi]_\pi, & \mathbf{T}[\pi\pi|\psi1]_\pi, & \mathbf{T}[\pi\pi|\psi\psi]_\pi, & \mathbf{T}[\pi\pi|\psi\pi]_\pi, & \mathbf{T}[\pi\pi|\pi1]_\pi, & \mathbf{T}[\pi\pi|\pi\psi]_\pi, \\ \mathbf{T}[\pi\pi|\pi\pi]_1, & \mathbf{T}[\pi\pi|\pi\pi]_\psi, & \mathbf{T}[\pi\pi|\pi\pi]_\pi \end{array} \right\}.$$

### Step 1.

As already hinted at above, the tube algebra decomposes into two 3-dimensional algebras and one 5-dimensional algebra: $M_3(\mathbb{C}) \oplus M_3(\mathbb{C}) \oplus M_5(\mathbb{C})$. A complete set of matrix units is given by

$$[e_1]_{ij} = \begin{pmatrix} \mathbf{T}[11|11]_1 & \mathbf{T}[\psi\psi|11]_\psi & \frac{\mathbf{T}[\pi\pi|11]_\pi}{\sqrt{2}} \\ \mathbf{T}[11|\psi\psi]_\psi & \mathbf{T}[\psi\psi|\psi\psi]_1 & \frac{\mathbf{T}[\pi\pi|\psi\psi]_\pi}{\sqrt{2}} \\ \frac{\mathbf{T}[11|\pi\pi]_\pi}{\sqrt{2}} & \frac{\mathbf{T}[\psi\psi|\pi\pi]_\pi}{\sqrt{2}} & \frac{1}{4}\left(\mathbf{T}[\pi\pi|\pi\pi]_1 + \mathbf{T}[\pi\pi|\pi\pi]_\psi + \sqrt{2}\,\mathbf{T}[\pi\pi|\pi\pi]_\pi\right) \end{pmatrix}_{ij} \tag{C2a}$$

$$[e_\psi]_{ij} = \begin{pmatrix} \mathbf{T}[1\psi|1\psi]_1 & \mathbf{T}[\psi1|1\psi]_\psi & \frac{\mathbf{T}[\pi\pi|1\psi]_\pi}{\sqrt{2}} \\ \mathbf{T}[1\psi|\psi1]_\psi & \mathbf{T}[\psi1|\psi1]_1 & \frac{\mathbf{T}[\pi\pi|\psi1]_\pi}{\sqrt{2}} \\ \frac{\mathbf{T}[1\psi|\pi\pi]_\pi}{\sqrt{2}} & \frac{\mathbf{T}[\psi1|\pi\pi]_\pi}{\sqrt{2}} & \frac{1}{4}\left(\mathbf{T}[\pi\pi|\pi\pi]_1 + \mathbf{T}[\pi\pi|\pi\pi]_\psi - \sqrt{2}\,\mathbf{T}[\pi\pi|\pi\pi]_\pi\right) \end{pmatrix}_{ij} \tag{C2b}$$

$$[e_\pi]_{ij} = \begin{pmatrix} \mathbf{T}[1\pi|1\pi]_1 & \mathbf{T}[\psi\pi|1\pi]_\psi & \mathbf{T}[\pi1|1\pi]_\pi & \mathbf{T}[\pi\psi|1\pi]_\pi & \frac{\mathbf{T}[\pi\pi|1\pi]_\pi}{\sqrt[4]{2}} \\ \mathbf{T}[1\pi|\psi\pi]_\psi & \mathbf{T}[\psi\pi|\psi\pi]_1 & \mathbf{T}[\pi1|\psi\pi]_\pi & \mathbf{T}[\pi\psi|\psi\pi]_\pi & -\frac{\mathbf{T}[\pi\pi|\psi\pi]_\pi}{\sqrt[4]{2}} \\ \mathbf{T}[1\pi|\pi1]_\pi & \mathbf{T}[\psi\pi|\pi1]_\pi & \mathbf{T}[\pi1|\pi1]_1 & \mathbf{T}[\pi\psi|\pi1]_\psi & \frac{\mathbf{T}[\pi\pi|\pi1]_\pi}{\sqrt[4]{2}} \\ \mathbf{T}[1\pi|\pi\psi]_\pi & \mathbf{T}[\psi\pi|\pi\psi]_\pi & \mathbf{T}[\pi1|\pi\psi]_\psi & \mathbf{T}[\pi\psi|\pi\psi]_1 & -\frac{\mathbf{T}[\pi\pi|\pi\psi]_\pi}{\sqrt[4]{2}} \\ \frac{\mathbf{T}[1\pi|\pi\pi]_\pi}{\sqrt[4]{2}} & -\frac{\mathbf{T}[\psi\pi|\pi\pi]_\pi}{\sqrt[4]{2}} & \frac{\mathbf{T}[\pi1|\pi\pi]_\pi}{\sqrt[4]{2}} & -\frac{\mathbf{T}[\pi\psi|\pi\pi]_\pi}{\sqrt[4]{2}} & \frac{1}{2}\left(\mathbf{T}[\pi\pi|\pi\pi]_1 - \mathbf{T}[\pi\pi|\pi\pi]_\psi\right) \end{pmatrix}_{ij}. \tag{C2c}$$

Note that we are redundantly using the labels $1, \psi, \pi$ for the simple objects in: $\mathbf{Rep}(S_3)$ as both a fusion and module category, and as labels for the irreducible representations of the tube category.

A basis for the representations is given by

$$[v_1]_i = [e_1]_{i0}, \qquad\qquad [v_\psi]_i = [e_\psi]_{i0}, \qquad\qquad [v_\pi]_i = [e_\pi]_{i0}, \tag{C3}$$

with

$$[e_x]_{i,j}[v_y]_k = \delta_x^y \delta_j^k [v_x]_i. \tag{C4}$$

These vectors have norm

$$\|[v_1]_i\| = \|[v_\psi]_i\| = 1 \qquad\qquad \|[v_\pi]_i\| = \sqrt{2}. \qquad (C5)$$

The fusion category $\mathcal{C}^*_{\mathcal{M}}$ therefore has 3 simple objects, labeled $1, \psi, \pi$. Note that we are labeling the objects in $\mathcal{C}^*_{\mathcal{M}}$ with the same labels as the input category $\mathcal{C}$ since, as shown below, $\mathcal{C}^*_{\mathcal{M}} \equiv \mathcal{C}$ as a fusion category. From this point, all labels are in $\mathcal{C}^*_{\mathcal{M}}$.

**Step 2.**

The composite basis is 683 dimensional. Unlike the two previous examples, $683 \neq 43^2$, since many of the composite pictures evaluate to 0. The tensor products of the representations $v_x$ form a 43 dimensional subspace with basis

$$\left\{ \begin{array}{l} [v_1]_0 \otimes [v_1]_0, \quad [v_1]_1 \otimes [v_1]_1, \quad [v_1]_2 \otimes [v_1]_2, \quad [v_\psi]_0 \otimes [v_\psi]_1, \quad [v_\psi]_1 \otimes [v_\psi]_0, \quad [v_\psi]_2 \otimes [v_\psi]_2, \\ [v_1]_0 \otimes [v_\psi]_0, \quad [v_1]_1 \otimes [v_\psi]_1, \quad [v_1]_2 \otimes [v_\psi]_2, \quad [v_\psi]_0 \otimes [v_1]_1, \quad [v_\psi]_1 \otimes [v_1]_0, \quad [v_\psi]_2 \otimes [v_1]_2, \\ [v_1]_0 \otimes [v_\pi]_0, \quad [v_1]_1 \otimes [v_\pi]_1, \quad [v_1]_2 \otimes [v_\pi]_2, \quad [v_1]_2 \otimes [v_\pi]_3, \quad [v_1]_2 \otimes [v_\pi]_4, \\ [v_\psi]_0 \otimes [v_\pi]_1, \quad [v_\psi]_1 \otimes [v_\pi]_0, \quad [v_\psi]_2 \otimes [v_\pi]_2, \quad [v_\psi]_2 \otimes [v_\pi]_3, \quad [v_\psi]_2 \otimes [v_\pi]_4, \\ [v_\pi]_0 \otimes [v_1]_2, \quad [v_\pi]_1 \otimes [v_1]_2, \quad [v_\pi]_2 \otimes [v_1]_0, \quad [v_\pi]_3 \otimes [v_1]_1, \quad [v_\pi]_4 \otimes [v_1]_2, \\ [v_\pi]_0 \otimes [v_\psi]_2, \quad [v_\pi]_1 \otimes [v_\psi]_2, \quad [v_\pi]_2 \otimes [v_\psi]_0, \quad [v_\pi]_3 \otimes [v_\psi]_1, \quad [v_\pi]_4 \otimes [v_\psi]_2, \\ [v_\pi]_0 \otimes [v_\pi]_2, \quad [v_\pi]_0 \otimes [v_\pi]_3, \quad [v_\pi]_0 \otimes [v_\pi]_4, \quad [v_\pi]_1 \otimes [v_\pi]_2, \quad [v_\pi]_1 \otimes [v_\pi]_3, \quad [v_\pi]_1 \otimes [v_\pi]_4, \\ [v_\pi]_2 \otimes [v_\pi]_0, \quad [v_\pi]_3 \otimes [v_\pi]_1, \quad [v_\pi]_4 \otimes [v_\pi]_2, \quad [v_\pi]_4 \otimes [v_\pi]_3, \quad [v_\pi]_4 \otimes [v_\pi]_4 \end{array} \right\}. \qquad (C6)$$

**Step 3.**

For the present example, there are no multiplicities. For clarity, we leave the free parameters unfixed, naming them $\omega$. This serves to demonstrate that they correspond to the gauge freedom in the $F$-symbols. The trivalent vertices are given by the matrices

$$\begin{array}{c} 1 \\ 1 \end{array} \!\!\!\succ\!\!\!- 1 = \begin{array}{c} \\ [v_1]_0 \otimes [v_1]_0 \\ [v_1]_1 \otimes [v_1]_1 \\ [v_1]_2 \otimes [v_1]_2 \end{array} \begin{array}{ccc} [v_1]_0 & [v_1]_1 & [v_1]_2 \\ \left( \omega^1_{11} \right. & 0 & 0 \\ 0 & \omega^1_{11} & 0 \\ 0 & 0 & \left. \sqrt{2}\omega^1_{11} \right) \end{array}$$

$$\begin{array}{c} \psi \\ 1 \end{array} \!\!\!\succ\!\!\!- \psi = \begin{array}{c} \\ [v_1]_0 \otimes [v_\psi]_0 \\ [v_1]_1 \otimes [v_\psi]_1 \\ [v_1]_2 \otimes [v_\psi]_2 \end{array} \begin{array}{ccc} [v_\psi]_0 & [v_\psi]_1 & [v_\psi]_2 \\ \left( \omega^\psi_{1\psi} \right. & 0 & 0 \\ 0 & \omega^\psi_{1\psi} & 0 \\ 0 & 0 & \left. \sqrt{2}\omega^\psi_{1\psi} \right) \end{array}$$

$$\begin{array}{c} 1 \\ \psi \end{array} \!\!\!\succ\!\!\!- \psi = \begin{array}{c} \\ [v_\psi]_0 \otimes [v_1]_1 \\ [v_\psi]_1 \otimes [v_1]_0 \\ [v_\psi]_2 \otimes [v_1]_2 \end{array} \begin{array}{ccc} [v_\psi]_0 & [v_\psi]_1 & [v_\psi]_2 \\ \left( \omega^\psi_{\psi 1} \right. & 0 & 0 \\ 0 & \omega^\psi_{\psi 1} & 0 \\ 0 & 0 & \left. \sqrt{2}\omega^\psi_{\psi 1} \right) \end{array}$$

$$\begin{array}{c} \psi \\ \psi \end{array} \!\!\!\succ\!\!\!- 1 = \begin{array}{c} \\ [v_\psi]_0 \otimes [v_\psi]_1 \\ [v_\psi]_1 \otimes [v_\psi]_0 \\ [v_\psi]_2 \otimes [v_\psi]_2 \end{array} \begin{array}{ccc} [v_1]_0 & [v_1]_1 & [v_1]_2 \\ \left( \omega^1_{\psi\psi} \right. & 0 & 0 \\ 0 & \omega^1_{\psi\psi} & 0 \\ 0 & 0 & \left. \sqrt{2}\omega^1_{\psi\psi} \right) \end{array}$$

$$
\begin{array}{c}
\pi \\
1
\end{array}\!\!\succ\!\!\pi \;=\;
\begin{array}{c}
\\
[v_1]_0 \otimes [v_\pi]_0 \\
[v_1]_1 \otimes [v_\pi]_1 \\
[v_1]_2 \otimes [v_\pi]_2 \\
[v_1]_2 \otimes [v_\pi]_3 \\
[v_1]_2 \otimes [v_\pi]_4
\end{array}
\begin{pmatrix}
[v_\pi]_0 & [v_\pi]_1 & [v_\pi]_2 & [v_\pi]_3 & [v_\pi]_4 \\
\omega^\pi_{1\pi} & 0 & 0 & 0 & 0 \\
0 & \omega^\pi_{1\pi} & 0 & 0 & 0 \\
0 & 0 & \sqrt{2}\omega^\pi_{1\pi} & 0 & 0 \\
0 & 0 & 0 & \sqrt{2}\omega^\pi_{1\pi} & 0 \\
0 & 0 & 0 & 0 & \sqrt{2}\omega^\pi_{1\pi}
\end{pmatrix}
$$

$$
\begin{array}{c}
1 \\
\pi
\end{array}\!\!\succ\!\!\pi \;=\;
\begin{array}{c}
\\
[v_\pi]_0 \otimes [v_1]_2 \\
[v_\pi]_1 \otimes [v_1]_2 \\
[v_\pi]_2 \otimes [v_1]_0 \\
[v_\pi]_3 \otimes [v_1]_1 \\
[v_\pi]_4 \otimes [v_1]_2
\end{array}
\begin{pmatrix}
[v_\pi]_0 & [v_\pi]_1 & [v_\pi]_2 & [v_\pi]_3 & [v_\pi]_4 \\
\sqrt{2}\omega^\pi_{\pi 1} & 0 & 0 & 0 & 0 \\
0 & \sqrt{2}\omega^\pi_{\pi 1} & 0 & 0 & 0 \\
0 & 0 & \omega^\pi_{\pi 1} & 0 & 0 \\
0 & 0 & 0 & \omega^\pi_{\pi 1} & 0 \\
0 & 0 & 0 & 0 & \sqrt{2}\omega^\pi_{\pi 1}
\end{pmatrix}
$$

$$
\begin{array}{c}
\pi \\
\psi
\end{array}\!\!\succ\!\!\pi \;=\;
\begin{array}{c}
\\
[v_\psi]_0 \otimes [v_\pi]_1 \\
[v_\psi]_1 \otimes [v_\pi]_0 \\
[v_\psi]_2 \otimes [v_\pi]_2 \\
[v_\psi]_2 \otimes [v_\pi]_3 \\
[v_\psi]_2 \otimes [v_\pi]_4
\end{array}
\begin{pmatrix}
[v_\pi]_0 & [v_\pi]_1 & [v_\pi]_2 & [v_\pi]_3 & [v_\pi]_4 \\
\omega^\pi_{\psi\pi} & 0 & 0 & 0 & 0 \\
0 & \omega^\pi_{\psi\pi} & 0 & 0 & 0 \\
0 & 0 & \sqrt{2}\omega^\pi_{\psi\pi} & 0 & 0 \\
0 & 0 & 0 & \sqrt{2}\omega^\pi_{\psi\pi} & 0 \\
0 & 0 & 0 & 0 & \sqrt{2}\omega^\pi_{\psi\pi}
\end{pmatrix}
$$

$$
\begin{array}{c}
\psi \\
\pi
\end{array}\!\!\succ\!\!\pi \;=\;
\begin{array}{c}
\\
[v_\pi]_0 \otimes [v_\psi]_2 \\
[v_\pi]_1 \otimes [v_\psi]_2 \\
[v_\pi]_2 \otimes [v_\psi]_0 \\
[v_\pi]_3 \otimes [v_\psi]_1 \\
[v_\pi]_4 \otimes [v_\psi]_2
\end{array}
\begin{pmatrix}
[v_\pi]_0 & [v_\pi]_1 & [v_\pi]_2 & [v_\pi]_3 & [v_\pi]_4 \\
\sqrt{2}\omega^\pi_{\pi\psi} & 0 & 0 & 0 & 0 \\
0 & \sqrt{2}\omega^\pi_{\pi\psi} & 0 & 0 & 0 \\
0 & 0 & 0 & \omega^\pi_{\pi\psi} & 0 \\
0 & 0 & \omega^\pi_{\pi\psi} & 0 & 0 \\
0 & 0 & 0 & 0 & -\sqrt{2}\omega^\pi_{\pi\psi}
\end{pmatrix}
$$

$$
\begin{array}{c}
\pi \\
\pi
\end{array}\!\!\succ\!\!1 \;=\;
\begin{array}{c}
\\
[v_\pi]_0 \otimes [v_\pi]_2 \\
[v_\pi]_0 \otimes [v_\pi]_3 \\
[v_\pi]_0 \otimes [v_\pi]_4 \\
[v_\pi]_1 \otimes [v_\pi]_2 \\
[v_\pi]_1 \otimes [v_\pi]_3 \\
[v_\pi]_1 \otimes [v_\pi]_4 \\
[v_\pi]_2 \otimes [v_\pi]_0 \\
[v_\pi]_3 \otimes [v_\pi]_1 \\
[v_\pi]_4 \otimes [v_\pi]_2 \\
[v_\pi]_4 \otimes [v_\pi]_3 \\
[v_\pi]_4 \otimes [v_\pi]_4
\end{array}
\begin{pmatrix}
[v_1]_0 & [v_1]_1 & [v_1]_2 \\
\omega^1_{\pi\pi}/\sqrt{2} & 0 & 0 \\
0 & 0 & 0 \\
0 & 0 & 0 \\
0 & 0 & 0 \\
0 & \omega^1_{\pi\pi}/\sqrt{2} & 0 \\
0 & 0 & 0 \\
0 & 0 & \omega^1_{\pi\pi}/4 \\
0 & 0 & \omega^1_{\pi\pi}/4 \\
0 & 0 & 0 \\
0 & 0 & 0 \\
0 & 0 & \omega^1_{\pi\pi}/2
\end{pmatrix}
$$

$$
\pi\!\!\diagdown\!\!\diagup\!\!\psi \;=\;
\begin{array}{c}
\phantom{[v_\pi]_0 \otimes [v_\pi]_2}\\
\end{array}
\begin{array}{r|ccc}
 & [v_\psi]_0 & [v_\psi]_1 & [v_\psi]_2 \\
\hline
[v_\pi]_0 \otimes [v_\pi]_2 & 0 & 0 & 0 \\
[v_\pi]_0 \otimes [v_\pi]_3 & \omega^\psi_{\pi\pi}/\sqrt{2} & 0 & 0 \\
[v_\pi]_0 \otimes [v_\pi]_4 & 0 & 0 & 0 \\
[v_\pi]_1 \otimes [v_\pi]_2 & 0 & \omega^\psi_{\pi\pi}/\sqrt{2} & 0 \\
[v_\pi]_1 \otimes [v_\pi]_3 & 0 & 0 & 0 \\
[v_\pi]_1 \otimes [v_\pi]_4 & 0 & 0 & 0 \\
[v_\pi]_2 \otimes [v_\pi]_0 & 0 & 0 & \omega^\psi_{\pi\pi}/4 \\
[v_\pi]_3 \otimes [v_\pi]_1 & 0 & 0 & \omega^\psi_{\pi\pi}/4 \\
[v_\pi]_4 \otimes [v_\pi]_2 & 0 & 0 & 0 \\
[v_\pi]_4 \otimes [v_\pi]_3 & 0 & 0 & 0 \\
[v_\pi]_4 \otimes [v_\pi]_4 & 0 & 0 & -\omega^\psi_{\pi\pi}/2
\end{array}
$$

$$
\pi\!\!\diagdown\!\!\diagup\!\!\pi \;=\;
\begin{array}{r|ccccc}
 & [v_\pi]_0 & [v_\pi]_1 & [v_\pi]_2 & [v_\pi]_3 & [v_\pi]_4 \\
\hline
[v_\pi]_0 \otimes [v_\pi]_2 & 0 & 0 & 0 & 0 & 0 \\
[v_\pi]_0 \otimes [v_\pi]_3 & 0 & 0 & 0 & 0 & 0 \\
[v_\pi]_0 \otimes [v_\pi]_4 & \omega^\pi_{\pi\pi} & 0 & 0 & 0 & 0 \\
[v_\pi]_1 \otimes [v_\pi]_2 & 0 & 0 & 0 & 0 & 0 \\
[v_\pi]_1 \otimes [v_\pi]_3 & 0 & 0 & 0 & 0 & 0 \\
[v_\pi]_1 \otimes [v_\pi]_4 & 0 & -\omega^\pi_{\pi\pi} & 0 & 0 & 0 \\
[v_\pi]_2 \otimes [v_\pi]_0 & 0 & 0 & 0 & 0 & \omega^\pi_{\pi\pi}/2 \\
[v_\pi]_3 \otimes [v_\pi]_1 & 0 & 0 & 0 & 0 & -\omega^\pi_{\pi\pi}/2 \\
[v_\pi]_4 \otimes [v_\pi]_2 & 0 & 0 & \omega^\pi_{\pi\pi} & 0 & 0 \\
[v_\pi]_4 \otimes [v_\pi]_3 & 0 & 0 & 0 & -\omega^\pi_{\pi\pi} & 0 \\
[v_\pi]_4 \otimes [v_\pi]_4 & 0 & 0 & 0 & 0 & 0
\end{array}\;.
$$

Note that these matrices are not isometric matrices, but are matrices for isometric operators.

## Step 4.

The full set of $F$-symbols is computed from these. Fixing $\omega^c_{ab}=1$, these are

$$
\begin{aligned}
&\left[F^1_{111}\right]_{11}=1 && \left[F^\psi_{11\psi}\right]_{1\psi}=1 && \left[F^\pi_{11\pi}\right]_{1\pi}=1 && \left[F^\psi_{1\psi1}\right]_{\psi\psi}=1 && \left[F^1_{1\psi\psi}\right]_{\psi1}=1 && \left[F^\pi_{1\psi\pi}\right]_{\psi\pi}=1 && \left[F^\pi_{1\pi1}\right]_{\pi\pi}=1 \\
&\left[F^1_{1\pi\psi}\right]_{\pi\pi}=1 && \left[F^1_{1\pi\pi}\right]_{\pi1}=1 && \left[F^\psi_{1\pi\pi}\right]_{\pi\psi}=1 && \left[F^\pi_{1\pi\pi}\right]_{\pi\pi}=1 && \left[F^\psi_{\psi11}\right]_{\psi1}=1 && \left[F^1_{\psi1\psi}\right]_{\psi\psi}=1 && \left[F^\pi_{\psi1\pi}\right]_{\psi\pi}=1 \\
&\left[F^1_{\psi\psi1}\right]_{1\psi}=1 && \left[F^\psi_{\psi\psi\psi}\right]_{11}=1 && \left[F^\pi_{\psi\psi\pi}\right]_{1\pi}=1 && \left[F^\pi_{\psi\pi1}\right]_{\pi\pi}=1 && \left[F^\pi_{\psi\pi\psi}\right]_{\pi\pi}=1 && \left[F^\psi_{\psi\pi\pi}\right]_{\pi1}=1 && \left[F^1_{\psi\pi\pi}\right]_{\pi\psi}=1 \\
&\left[F^\pi_{\psi\pi\pi}\right]_{\pi\pi}=-1 && \left[F^\pi_{\pi11}\right]_{\pi1}=1 && \left[F^\pi_{\pi1\psi}\right]_{\pi\psi}=1 && \left[F^1_{\pi1\pi}\right]_{\pi\pi}=1 && \left[F^\psi_{\pi1\pi}\right]_{\pi\pi}=1 && \left[F^\pi_{\pi1\pi}\right]_{\pi\pi}=1 && \left[F^\pi_{\pi\psi1}\right]_{\pi\psi}=1 \;,\\
&\left[F^\pi_{\pi\psi\psi}\right]_{\pi1}=1 && \left[F^1_{\pi\psi\pi}\right]_{\pi\pi}=1 && \left[F^\pi_{\pi\psi\pi}\right]_{\pi\pi}=1 && \left[F^\pi_{\pi\psi\pi}\right]_{\pi\pi}=-1 && \left[F^1_{\pi\pi1}\right]_{1\pi}=1 && \left[F^\psi_{\pi\pi1}\right]_{\psi\pi}=1 && \left[F^\pi_{\pi\pi1}\right]_{\pi\pi}=1 \\
&\left[F^\psi_{\pi\pi\psi}\right]_{1\pi}=1 && \left[F^1_{\pi\pi\psi}\right]_{\psi\pi}=1 && \left[F^\pi_{\pi\pi\psi}\right]_{\pi\pi}=-1 && \left[F^\pi_{\pi\pi\pi}\right]_{11}=\tfrac{1}{2} && \left[F^\pi_{\pi\pi\pi}\right]_{1\psi}=\tfrac{1}{2} && \left[F^\pi_{\pi\pi\pi}\right]_{1\pi}=\tfrac{1}{\sqrt{2}} && \left[F^\pi_{\pi\pi\pi}\right]_{\psi1}=\tfrac{1}{2} \\
&\left[F^\pi_{\pi\pi\pi}\right]_{\psi\psi}=\tfrac{1}{2} && \left[F^\pi_{\pi\pi\pi}\right]_{\psi\pi}=-\tfrac{1}{\sqrt{2}} && \left[F^\pi_{\pi\pi\pi}\right]_{\pi1}=\tfrac{1}{\sqrt{2}} && \left[F^\pi_{\pi\pi\pi}\right]_{\pi\psi}=-\tfrac{1}{\sqrt{2}} && \left[F^1_{\pi\pi\pi}\right]_{\pi\pi}=1 && \left[F^\psi_{\pi\pi\pi}\right]_{\pi\pi}=-1 && \left[F^\pi_{\pi\pi\pi}\right]_{\pi\pi}=0
\end{aligned}
$$

the $F$-symbols for $\mathbf{Rep}(S_3)$ as expected.