# Peer review of "Computing associators of endomorphism fusion categories"

_SciPost Physics_

## Round 2 · Referee Report · Ana Ros Camacho (Referee 1) · 2022-2-24

Strengths

1-The paper provides a new computational pathway to describing the associators of the whole categorical Morita equivalence class through just one representative of it. 2-The examples provided to show the power of this algorithm, in particular the Haagerup fusion categories, are really nice and exciting! 3- The paper ticks all the boxes for the acceptance criteria: it is well-written and has a clear structure. It includes examples of the algorithm described which are interesting. It has a detailed abstract, and a good summary of achievements. Citations included are of top quality. Files and code are available and nice to read.

Weaknesses

1-Through the manuscript the authors use (indecomposable) module categories extensively. These are actually tricky to describe in general, even when skeletizing, which makes me hesitate about the actual use of the algorithm described beyond some known cases. Still, the examples described and in particular the one about the Haagerup fusion categories are exciting ones. 2-The paper lacks a bit of perspective for future work - some comments on this would benefit the manuscript quite a bit. 3- The manuscript needs some (minor) clarifications here and there, and also about notation. E.g.: -- the simples of the skeletal fusion category in Definition 1 are denoted as $a_1$, $a_2$, etc but this notation is not used again later (similarly with the simples of the module category). -- Eq 5a: the bullet notation at the vertex has not been introduced at this point; also a word about what $M_{ab}^c$ is at this point would help. -- Right before Definition 3 the notion of indecomposable module category is mentioned but this hasn't been introduced earlier (and this is quite a crucial concept for the construction described, so maybe worth a sentence!). -- Speaking of which, I would include some citation on the tube algebra, in particular when claiming that when a module category is indecomposable the tube algebra is semisimple since it's an important point. -- At the beginning of the unitary subsection at section II, it is unclear to me what the * notation means (since it is used earlier at Eq 18). -- The F introduced right before Eq 38, is it the same F or is it anything new? 4-I would like to ask the authors for a justification of the software used in the algorithm described. Mathematica is notoriously known for e.g. omitting certain solutions when solving equations, which may lead to wrong statements. Why this software is a good one, and why it is not interfering with the results obtained? 5-While the strategy performed and the algorithm are interesting I believe the paper lacks a bit of mathematical depth, which is not necessarily a bad thing for this journal.

Report

I recommend this article for publication at SciPost Physics. Well done folks!

Requested changes

Please could you address points 2, 3 and 4 from the "Weaknesses" part.

  • validity: top
  • significance: high
  • originality: high
  • clarity: top
  • formatting: perfect
  • grammar: excellent

Author:  Jacob C Bridgeman  on 2022-04-25  [id 2414]

(in reply to Report 1 by Ana Ros Camacho on 2022-02-24)
Category:
answer to question

We thank the referee for their time and recommendation of our paper for publication at SciPost Physics. We address the weaknesses below. Additionally, we provide a pdf with changed marked.

  1. We have added some suggestions for future work in Section VI.

  2. We have tweaked the manuscript to address each of these comments.

  3. We have used Mathematica since it allows easy access to exact solutions via the 'Root' symbol. For our purposes here, omission of solutions is not a major concern. We only ask Mathematica to solve equations at two steps: When finding the matrix units (step 1), we can verify the solution by checking they span the appropriate matrix algebra. When solving for the dual F-symbols, the solution is unique, so omission would correspond to no solution being found.

Attachment:

ComputingFsEndoCats.pdf

---

## Round 2 · Referee Report · Anonymous (Referee 2) · 2022-4-17

Report

Dear editor and authors,

First of all, my apologies for the long delay in my report. The delay does not reflect lack of interest in this manuscript.

The authors present a way to calculate the F-symbols associated with a particular fusion category. The starting point are the F-symbols of a different, but related fusion category, which are assumed to be known. The relation between the two fusion categories is that they are 'Morita equivalent'. In short, the idea is to construct the so-called module category over the fusion category with known F-symbols, and considering the module tube category, which allows for the calculation of the F-symbols associated with the fusion category one started with.

My general impression of the paper is that it is well written, and accessible for people with knowledge about fusion- and modular tensor categories, even if they are not too familiar with module- and module tube categories, which lies at the heart of the construction. In this respect, the explicitly worked out examples are really helpful and perhaps essential.

In the introduction, the authors point out the complexity of obtaining the F-symbols for a given set of fusion rules, which lies in the complexity of the (typically enormously overdetermined) consistency conditions, the pentagon equations. The large amount of gauge symmetry that needs to be fixed adds to the problem, in particular in the presence of fusion multiplicities. The algorithm the authors present is really helpful (if available), because the equations that need to be solved are typically much simpler. Apart from the simpler examples, the authors also apply the algorithm, to find the F-symbols of H_1 (coming from the Haagerup subfactor), which were previously unknown.

In section II, the authors provide the necessary background information, starting with some details on fusion categories, C-module categories, module tube categories and Morita equivalence.

In section III, the authors use the notions reviewed in section II, to calculate the data for the module category, in particular the decomposition of the tensor products of pairs of irreducible representations, which can be used in the end to calculate the sought after F-symbols.

Section IV contains a relatively simple example that is worked out in detail, to show the workings of the algorithm, while section V contains the more complicated case of H_1 I alluded to above.

In my opinion, this is an interesting, technical but well written paper on the subject of fusion categories. In my opinion, this paper should be published in scipost. Below, I provide a list of comments, questions, recommendations, etc., which I urge the authors to consider.

Requested changes

Introduction

1. It's presumably good to point out that the construction presented in the paper gives a particular solution for the F-symbols associated with a set of fusion rules, but typically not all solutions (this is the usual situation, so no criticism to the paper).

Some related questions. If one has several monoidially inequivalent sets of F-symbols for the Morita equivalent fusion category, are the resulting sets of F-symbols also monoidially inequivalent? If one happens to know all monoidially inequivalent sets of F-symbols for the Morita equivalent fusion category, does one obtain all monoidially inequivalent sets of F-symbols for the fusion category of interest?

Section II

2. I am confused about the discussion of the Frobenius-Perron dimension (around eq. (7)). The Frobenius-Perron dimension of a is given by the largest eigenvalue of the fusion matrix N_a. As such, one has d_a >= 1, and satisfies 7b. However, it can not always be written as in the last equation of eq. (6). In non-unitary fusion categories, the F-symbol appearing in that equation can have absolute value larger than one, in contradiction with d_a >= 1. So in some way, it looks like that with d_a, the authors mean the quantum dimension. However, in non-unitary fusion categories, there will be negative quantum dimension, so d_a can not be the quantum dimension either.

3. A somewhat related issue to question 2. Just under eqns (7), it is stated that if there is a basis, such that all F-matrices are unitary, the fusion category is (or is called) unitary. This is strictly speaking not true (though for perhaps a rather trivial reason). Even if all F-matrices are unitary, it is sometimes possible to choose a pivotal structure, such that some of the quantum dimensions are negative. One would not call such a fusion category unitary.

  1. For the C-module category, I am wondering if the same issues arise as under 2. and 3. for the fusion category.

Section III

  1. The notation in eq. (30) is not entirely clear/defined. Are the C^{\alpha}_P simply coefficients that need to be found?

Section IV

  1. The first 1/4 in eq. (47a) should be 1/2.

Section V.

  1. It is stated that it s considered likely that a CFT associated with the Haagerup subfactor exists. I am wondering what data of this conjectured (?) CFT is known. It could be interesting to use the F-symbols obtained for H_1, and solve the hexagon equations (which is typically much easier than solving the pentagon equations). In this way one could obtain information about the scaling dimensions of the primary fields (or compare, if this information is available).

  2. Here, the authors deal with the Haagerup fusion categories. They provide a visualisation, which is of course useful. It is however not stated what it is the authors are plotting. Without this information, it is hard to learn something from the figure.

Generic question

  1. In the paper, the authors work out three examples in detail (Sec. IV, App. B, C). Do these three examples cover all possible complications that can arise when executing the algorithm? In either case, it would be good to point this out somewhere.

  • validity: high
  • significance: good
  • originality: high
  • clarity: good
  • formatting: excellent
  • grammar: excellent

Author:  Jacob C Bridgeman  on 2022-04-25  [id 2415]

(in reply to Report 2 on 2022-04-17)
Category:
remark
answer to question

We thank the referee for their close reading of our manuscript. We address each of the requested changes below. Additionally, we provide a pdf with changed marked.

Introduction:

1.

We are a little confused by this comment. We indicate several times that we compute the $F$-symbols of the Morita equivalent category $C_M^*$, which are uniquely determined (up to tensor equivalence/gauge+permutation)/

Section II:

2.

We have rearranged this section, introducing a heading 'Unitary case' to clarify. Since we only need to restrict the gauge in this way in the unitary case, we have moved the discussion of FP dimensions to that section.

3.

We have rearranged the sentence to recognize that the unitarity of $F$ is a consequence of unitarity of the category.

4.

We don't think this should be an issue, but we've added a footnote indicating that the pivotal structure should be the one compatible with unitarity.

Section III:

5.

We have clarified this.

Section IV:

6.

We have corrected this.

Section V:

  1. The data for the conjectured CFT was worked out by Evans and Gannon in https://arxiv.org/abs/1006.1326. They showed that the central charge of the CFT is a multiple of 8, and construct some character vectors for the corresponding vertex operator algebra. However, this data corresponds to the CFT associated with the quantum double of the Haagerup fusion categories. The problem with the fusion categories themselves is that the respective hexagon equations do not have a solution, so none of them admits a braiding.

7.

We have clarified what the figure shows.

Generic question:

8.

We've added an appropriate comment just above Section V

Attachment:

ComputingFsEndoCats_zUS1ZUB.pdf

---

## Editorial Decision

resubmitted